# Abiraterone acetate preferentially enriches for the gut commensal *Akkermansia muciniphila* in castrate-resistant prostate cancer patients

Brendan A. Daisley [1,2,3], Ryan M. Chanyi [1,2,3], Kamilah Abdur-Rashid[1,2,3], Kait F. Al [1,2,3], Shaeley Gibbons[1,2,3], John A. Chmiel [1,2,3], Hannah Wilcox[1,2,3], Gregor Reid [1,2,3], Amanda Anderson[4], Malcolm Dewar[4], Shiva M. Nair [4], Joseph Chin[4] & Jeremy P. Burton [1,2,3,4]✉

Abiraterone acetate (AA) is an inhibitor of androgen biosynthesis, though this cannot fully explain its efficacy against androgen-independent prostate cancer. Here, we demonstrate that androgen deprivation therapy depletes androgen-utilizing *Corynebacterium* spp. in prostate cancer patients and that oral AA further enriches for the health-associated commensal, *Akkermansia muciniphila*. Functional inferencing elucidates a coinciding increase in bacterial biosynthesis of vitamin K2 (an inhibitor of androgen dependent and independent tumor growth). These results are highly reproducible in a host-free gut model, excluding the possibility of immune involvement. Further investigation reveals that AA is metabolized by bacteria in vitro and that breakdown components selectively impact growth. We conclude that *A. muciniphila* is a key regulator of AA-mediated restructuring of microbial communities, and that this species may affect treatment response in castrate-resistant cohorts. Ongoing initiatives aimed at modulating the colonic microbiota of cancer patients may consider targeted delivery of poorly absorbed selective bacterial growth agents.

[1] Department of Microbiology and Immunology, The University of Western Ontario, London, ON N6A 5C1, Canada. [2] Canadian Centre for Human Microbiome and Probiotics Research, London, ON N6C 2R5, Canada. [3] Lawson Health Research Institute, St. Joseph's Health Care London, London, ON N6A 4V2, Canada. [4] Department of Surgery, Division of Urology, Schulich School of Medicine, London, ON N6A 5C1, Canada. ✉email: jeremy.burton@lawsonresearch.com

Several recent reports have demonstrated that the human microbiome plays a critical role in cancer development and progression, as well as treatment efficacy[1–3]. One bacterial species, *Akkermansia muciniphila*, appears to be particularly important for positive response to anti-PD-1-based immunotherapies. This species has also been inversely correlated with inflammation, obesity, and a variety of metabolic disorders in mice and humans[4,5]. However, there remains limited knowledge on how the gastrointestinal (GI) microbiota affects distant malignancies, such as prostate cancer (PC). This disease affects approximately one in seven men, has high morbidity rates, and is a tremendous financial burden on healthcare systems. The progression of PC relies on androgen receptor activation, primarily by testosterone and dihydrotestosterone. In PC patients with metastatic or high-risk localized disease, androgen deprivation therapy (ADT) via medical castration is a key part of therapy administered in the form of systemic (intramuscular or subcutaneous) gonadotropin-releasing hormone agonists, such as goserelin, leuprorelin, or triptorelin. Disease progression despite castrate levels of androgens is termed castrate-resistant PC and is commonly treated with oral androgen receptor axis-targeted therapies, such as abiraterone acetate (AA). Notably, AA is poorly absorbed, and it is estimated that ~55% of the unaltered parent compound and 22% of abiraterone from an administered dose are excreted in the feces[6]—demonstrating that the patient GI microbiota is exposed to high concentrations of AA and that partial metabolism of AA may occur in the gut.

Most studies to date have not considered the direct influence of PC treatments on the GI microbiota or vice versa. Based on the known association between serum sex steroid hormone levels and GI microbiota composition[7], we suspected that androgen depletion in PC patients receiving ADT might have unintentional consequences on microbial homeostasis and thereby impact clinical outcome. We also theorized that xenobiotic metabolism of AA by the GI microbiota of PC patients could be an unintended consequence of oral delivery, but a likely scenario based on previous studies showing microbiota-mediated biotransformation of irinotecan, non-steroidal anti-inflammatory drugs, digoxin, metformin, and a myriad of other pharmaceutical drugs[8].

In this study, we evaluate how systemic ADT and oral AA treatments differentially impact the GI microbiota of PC patients. The findings demonstrate that systemic depletion of androgens via ADT also depletes androgen-utilizing and pro-inflammatory *Corynebacterium* spp. Alternatively, oral AA distinctively remodels the GI microbiota characterized most notably by promotion of the health-associated and anti-inflammatory gut commensal, *A. muciniphila*. Functional inferencing suggests that these changes are also associated with a shift towards increased bacterial biosynthesis of certain host vitamins relevant to PC. Through further investigation and modeling of this unique microbiota–xenobiotic interaction in a host-free environment, we identify that *A. muciniphila* is a major determinant influencing the outcome of AA exposure on human-derived bacterial communities—a clinically relevant finding suggests that *A. muciniphila* may improve oral AA treatment response in PC patients.

## Results

**PC treatment correlates with patient microbiota**. To investigate potential interactions between PC treatment drugs and the GI microbiota, we performed 16S rRNA gene sequencing on fecal samples collected from 68 PC patients following routine digital rectal examination (patient characteristics outlined in Supplementary Table 1). The rectal microbiota composition of those receiving either ADT or ADT combined with oral AA (ADT + AA) was significantly different based on several measures of comparison relative to control patients not receiving any form of therapy (Fig. 1). Summarizing the dataset, a principal component analysis (PCA) exploring patient microbiota differences (based on Aitchison distances between samples) demonstrated that the microbiota composition of ADT and ADT + AA patients shifted slightly from that of the control group along the PC1 axis and each other along the PC2 axis— accounting for 20.4% and 15.7% of interpersonal microbiota variability, respectively (Fig. 1a). The two largest influencers driving these directional shifts, based on genus-level ordination, were found to be *Akkermansia* and *Corynebacterium* (Fig. 1b, c). Additional overview analyses were performed on the basis of previously reported differences in GI microbiota alpha diversity between PC patients and healthy controls[9]. Shannon's diversity (a metric of bacterial community alpha diversity accounting for species abundance and evenness) was found to be significantly lower in ADT + AA samples compared to control samples (Supplementary Fig. 1A–C).

Given that a variety of pharmaceutical drugs and various disease states are known to influence the GI microbiota, we examined a broad range of patient variables during evaluation of potential confounders in the clinical metadata (Supplementary Table 1). Out of the 17 total factors evaluated, only AA ($p = 0.001$), ADT ($p = 0.043$), and corticosteroid treatment ($p = 0.049$) had a significant impact on overall microbiota composition (Supplementary Table 2). We concluded from this that no patient variables better explained the variance in microbiota composition than the PC treatments themselves and that identification of corticosteroids as having a significant impact on microbiota composition was likely due to the disproportionally high degree of overlap with prescribed regimens including ADT.

**Androgen deprivation reduces androgen-utilizing gut bacteria**. In comparing differentially abundant taxa between patient groups, ADT and ADT + AA samples both demonstrated a decreased relative abundance in *Corynebacterium* (95% confidence intervals [CIs], 0.001–0.064% and 0.001–0.099%, relative abundance, respectively) compared to the control group (95% CI, 0.097–0.509% relative abundance). *Corynebacterium* spp. belong to the phylum Actinobacteria and are well-known testosterone metabolizers (some of which possess 5α-reductase activity[10]) and have been shown to colonize androgen-rich anatomical sites of males, including the axilla[11] and urogenital tract[12]. Exploratory inquiry demonstrated a general trend towards a decreased proportion of several other steroid-metabolizing genera within Actinobacteria in either ADT, ADT + AA, or both treatment groups (Fig. 1d, e and Supplementary Data 1). These results suggest that depletion of circulating androgen levels via systemic ADT may indirectly reduce host colonization by testosterone-metabolizing species. This substantiates previous findings on the interplay between serum sex steroid hormone levels and the GI microbiota[7].

In addition, we demonstrate that AA can directly inhibit several testosterone-metabolizing Actinobacteria in vitro, including multiple isolates of human-derived *Corynebacterium* spp. (Supplementary Fig. 2E). We confirmed that this inhibition was due to the abiraterone portion of AA and that acetate alone was unable to inhibit any of the strains tested (Supplementary Fig. 2F–H). Alongside the structural similarities and interkingdom receptor activation of various steroid hormones and bacterial autoinducers[13], our findings suggest that the active portion of AA (abiraterone; a steroidal progesterone derivative) may interfere with bacterial growth dynamics. One potential mechanism could be cross-reactivity of AA with actinobacterial steroid hydroxylases that, like mammalian CYP17A, are capable of catalyzing carbon–carbon bond cleavages[14]. While AA is

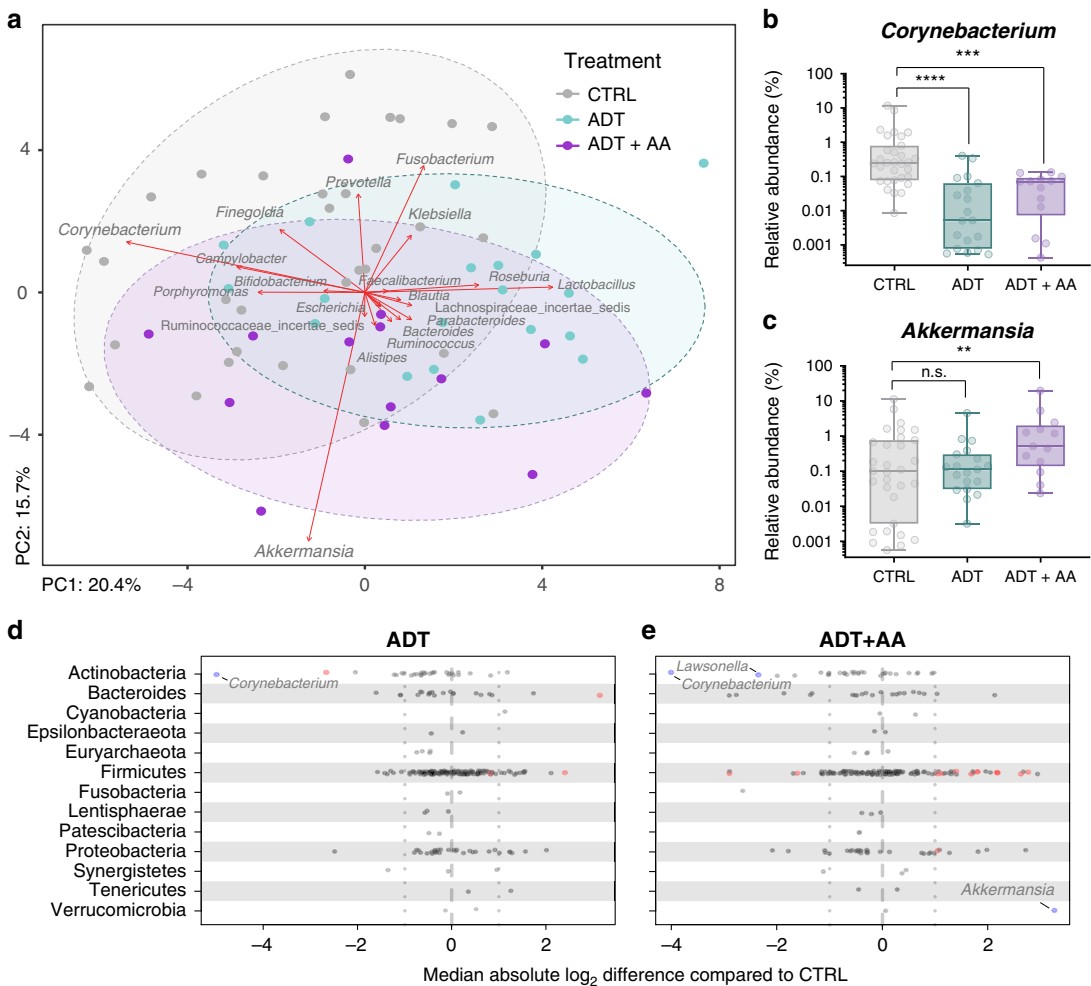

**Fig. 1 Rectal swab microbiota from prostate cancer patients receiving no treatment CTRL, ADT, or ADT + AA. a** Principal component analysis (PCA) plot of the microbiota from patient samples. Sequence variants were collapsed at genus-level identification, with clr-transformed Aitchison distances used as input values for PCA analysis. The distance between points represent differences in microbiota composition. Strength of association for taxa are depicted by the length of red arrows shown. Ellipses indicate 95% confidence intervals for each group. **b**, **c** Percent relative abundance of the two largest influencers of microbiota separation based on treatment. Data represent the median (line in box), IQR (box), and minimum/maximum (whiskers) for CTRL ($n = 33$), ADT ($n = 21$), and ADT + AA ($n = 14$) patient samples. Statistics shown are derived from multivariate analysis performed using MaAsLin2 additive generalized linear models with $\log_{10}$ clr-transformed input values. **d**, **e** ALDEx2 strip charts showing differential abundances of taxa between different patient groups. Positive values indicate increased relative abundance, while negative values indicate decreased relative abundance in the specified treatment groups (ADT or ADT + AA) relative to the CTRL samples. Statistical analysis performed with ALDEx2 and MaAsLin2 software. Features are colored red if ALDEx2 effect size differences (>1) and MaAsLin2 $p$ value (<0.05) thresholds are exceeded, and blue if effect size difference (>2) and $p$ value (<0.05) thresholds are exceeded. **P = 0.0030, ***p = 0.0007, ****p < 0.0001, and n.s. = not significant.

highly selective towards human enzymes, its potential effect on microbial enzymes in the gut warrants further investigation.

***Akkermansia muciniphila* modulates AA effects on patient microbiota**. In contrast to the depletion of Actinobacteria, a considerable enrichment of *A. muciniphila* was observed in ADT + AA patient samples (95% CI = 0.094–2.436%) compared to both ADT (95% CI = 0.031–0.304%) and no treatment control (95% CI = 0.035–0.565%) group samples (Fig. 1e). A recent smaller size study ($n = 30$) also demonstrated *A. muciniphila* to be increased in patients receiving oral androgen receptor axis-targeted therapies such as AA, bicalutamide, and enzalutamide[9]. However, the authors did not differentiate between the three oral treatments in their statistical analyses, making it difficult to ascertain whether the observed effects were due to host-mediated hormone depletion, direct drug interactions with the gut micro-biota, or both.

To delineate between host hormone–microbe interactions and potential immune-based or cytochrome P450 inhibition-mediated alterations to the gut microbiota, we investigated the direct effects of AA on fecal microbiota dynamics in vitro. A standardized amount of fecal inoculum was prepared from eight donors not receiving any form of PC treatment and then grown anaerobically in brain heart infusion (BHI) media supplemented with either AA or vehicle. Following 24 h of growth, we found a substantial decrease in beta diversity (i.e., AA-exposed samples had more similar microbiota compositions than did non-exposed samples) when *A. muciniphila* was present at a relative abundance of 0.1% or higher, whereas no change was observed in samples with low *A. muciniphila* levels (<0.1% relative abundance; Fig. 2a–c). Similarly, alpha diversity of AA-exposed samples also demonstrated a similar trend showing that the degree to which AA altered microbial diversity indices was dependent on background levels of *A. muciniphila* in patient

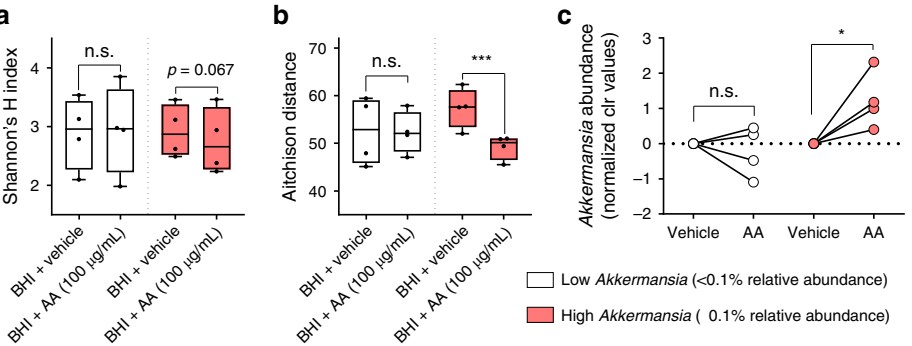

**Fig. 2 In vitro incubation of patient fecal samples with abiraterone acetate (AA).** Freshly collected fecal samples from donors not receiving any form of prostate cancer treatment ($n = 8$) were transferred to BHI media supplemented with 100 μg/mL of abiraterone acetate (AA) or vehicle (EtOH). Samples were incubated anaerobically at 37 °C for 48 h prior to 16S rRNA gene sequencing. **a** Alpha diversity was measured via Shannon's $H$ index and **b** beta diversity was measured via Aitchison distance between samples within the same group. Data represent the median (line in box), IQR (box), and minimum/maximum (whiskers) of $n = 4$ low *Akkermansia* and $n = 4$ high *Akkermansia* samples. Statistical comparisons shown for separate Wilcoxon's matched-pairs tests with multiple comparisons corrected using the Benjamini–Hochberg FDR method. **c** Differences in *Akkermansia* abundances following incubation with AA compared to vehicle. Data shown represent $\log_{10}$ clr-transformed relative abundances normalized to the vehicle group for each sample. Statistics shown are derived from differential abundance analysis using ALDEx2 software in R. \*\*\*$P = 0.0005$, \*$p = 0.0358$, and n.s. = not significant.

samples. These findings are clinically relevant and show that *A. muciniphila* is capable of modulating a specific shift in bacterial community dynamics in response to AA. Given that previous literature has positively correlated *A. muciniphila* levels in the gut with cancer treatment success[1–3], we chose to investigate this species further in hopes of elucidating potential ways to enhance the therapeutic efficacy of AA.

**Effects of AA are reproducible in a host-free gut model.** Next, to deconvolute the complexity of deciphering microbe–drug interactions over time in the human body, we utilized a simulated model of the human distal gut microbiota to simulate AA exposure in the context of a stabilized polymicrobial community within a host-free environment (Fig. 3). Following community stabilization, we exposed the simulated gut model to AA (250 mg/day) for six consecutive days. Initial AA exposure led to a stark increase in *A. muciniphila* levels (>130-fold increase in relative abundance within 24 h post AA exposure) that persisted over the exposure period (peaking at >1000-fold increase in relative abundance) and gradually decreased following AA withdrawal (Fig. 3b). We performed a PCA on time-course collected samples from the gut model (at genus-level ordination) and identified that samples from the AA exposure period separated from the non-exposure period on PC1 axis (22.6% explained variance), while all samples indiscriminatory of AA exposure were spread out along the PC2 axis (56.4% explained variance) in what appeared to be a temporally driven drift in composition (Fig. 3c).

After adjusting for time as a confounder, gut model samples from AA exposure days were found to be enriched with *Akkermansia*, *Klebsiella*, *Bilophila*, *Hungatella*, and *Oscillibacter* (Supplementary Table 3). Alternatively, AA exposure led to a depletion in *Faecalibacterium*, *Coprococcus*, and unclassified Lachnospiraceae group 3007. Corroborating these findings, a co-occurrence network analysis of the gut model samples also showed that AA exposure was positively correlated with *Akkermansia* (which itself was connected to several important hub genera) and negatively correlated very strongly with *Faecalibacterium*, *Coprococcus*, and unclassified Lachnospiraceae group ND3007 (Fig. 3f). *Faecalibacterium* and *Coprococcus* specifically showed a near opposite response to AA compared to *A. muciniphila* (Supplementary Fig. 3), which is interesting given that they are both well-known butyrate producers[15] and that *F. prausnitzii* has been shown to positively interact with *A.*

*muciniphila* on multiple occassion[16,17]; future studies would benefit from investigating the mechanism by which AA negatively impacts butyrate-producing bacteria. These findings overlap considerably with the ADT + AA patient microbiota dataset and solidify the idea that AA can alter human microbial communities independent of host factors (Supplementary Data 1, Supplementary Table 3, Fig. 3a–d, and Supplementary Fig. 4). Moreover, preferential growth status of *A. muciniphila* during early AA exposure in the host-free gut model further supported its chief regulatory role in the facilitation of AA-mediated modulation of microbial communities.

**AA uniquely promotes *A. muciniphila* growth in pure culture.** Next, we assessed bacterial metabolism of AA by growing *A. muciniphila* and 17 other bacterial strains of interest (mostly Enterobacteriaceae isolates showing a capacity to utilize AA as a sole carbon sole; Supplementary Fig. 2E, Supplementary Fig. 5A–E, and Supplementary Table 4) in media supplemented with AA and then measured breakdown in culture supernatants after 24 h growth. High-performance liquid chromatography (HPLC) analysis confirmed that all strains tested were able to breakdown ~70% of the AA present relative to the uninoculated media control (Fig. 3e). However, subsequent growth assays demonstrated that only *A. muciniphila* could derive a direct growth advantage from AA in pure culture alongside other carbon sources, while all other strains showed either no effect or a decreased growth potential (Fig. 3g, Supplementary Fig. 2, and Supplementary Fig. 5). Supplementation of growth media with sodium acetate at equimolar concentration elicited a near-identical growth response in *A. muciniphila*, suggesting that the acetate portion of AA was the responsible factor mediating enhanced growth (Fig. 3h, i).

**AA exposure correlates with predicted bacterial vitamin biosynthesis.** Based on the evidence so far suggesting that *A. muciniphila* is privileged in receiving a growth benefit from AA, we performed functional inferencing on 16S rRNA gene sequencing datasets from PC patients and the simulated gut model to determine how these interactions might impact overall bacterial metagenome potential (Fig. 4). We found that out of 306 total MetaCyc pathways predicted, glyoxylate cycle-related pathways (which permit the bypassing of two decarboxylation steps in the citric acid cycle and allow bacteria to grow on acetate via the

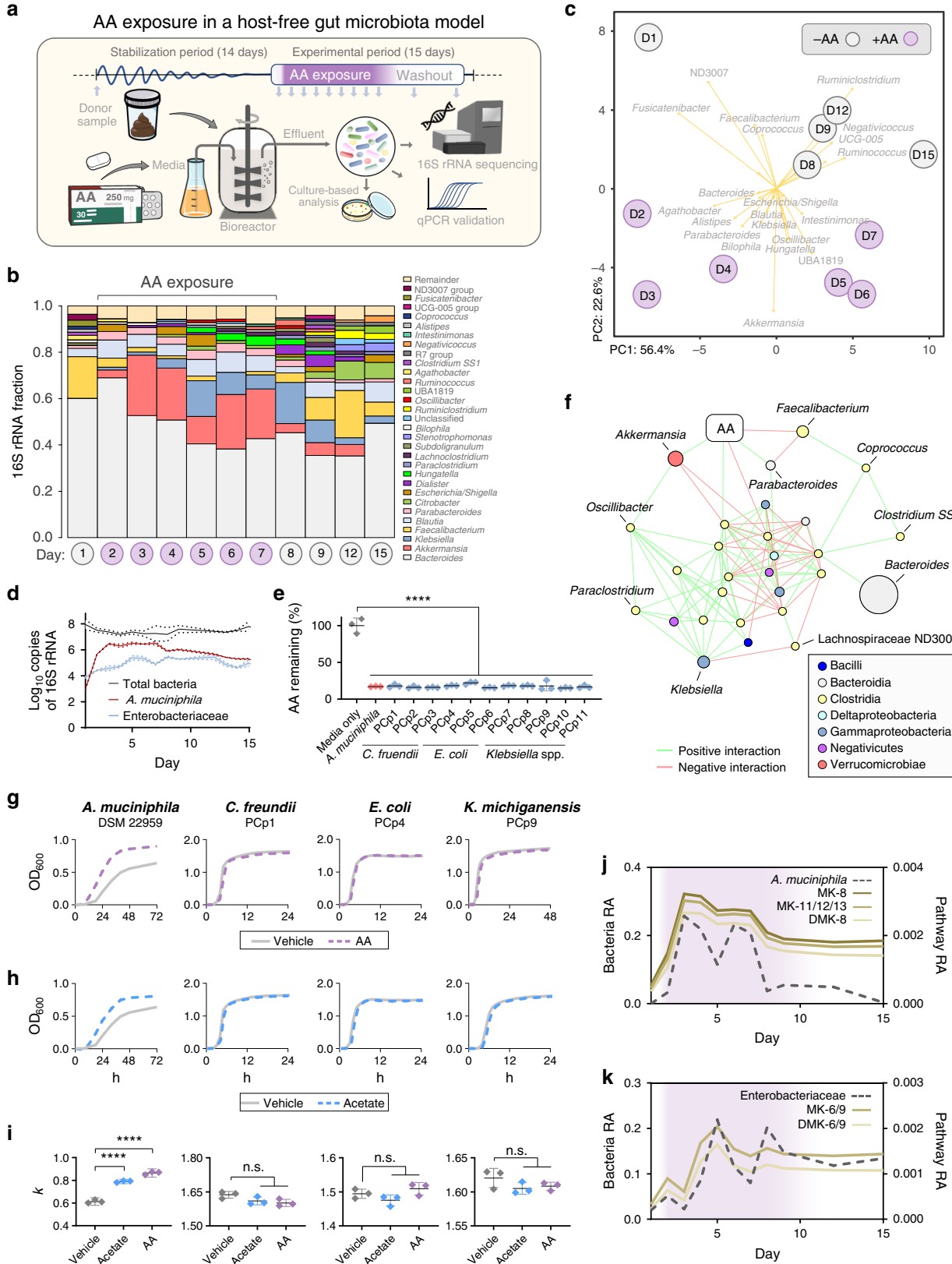

direct conversion to succinate[18]) and nearly all menaquinone (MKn, where $n$ = sidechain prenyl subunit length; collectively referred to vitamin K2) biosynthesis-related pathways were consistently increased in both AA-exposed gut model samples and ADT + AA patient samples compared to relative controls (Fig. 4a–i, Supplementary Fig. 6A, B, Supplementary Fig. 7,

Supplementary Table 5, and Supplementary Data 2–6). In addition, four pathways related to mycolic acid biosynthesis were nearly undetectable in both ADT and ADT + AA patient samples compared to controls (Supplementary Fig. 8).

Quinones (which can act as lipid-soluble electron carriers) have recently been shown to facilitate syntrophic growth via

**Fig. 3 AA exposure promotes *A. muciniphila* in a simulated model of the human distal gut microbiota. a** Simplified schematic providing a detailed overview of experimental methodology. Servier Medical Art images were used and modified under the Creative Commons Attribution 3.0 Unported License. **b** Bar plot representing the microbiota compositions of gut model samples from before, during, and after AA exposure as determined by sequencing of the V4 region of the bacterial 16S rRNA gene. **c** PCA plot of time-course collected gut model samples matching the days shown. Aitchison distance of genus-level microbiota compositions were used as input values and strength of association for taxa are depicted by the length of arrows shown. **d** qPCR-based quantification of total bacteria, Enterobacteriaceae, and *A. muciniphila* in the gut model over time. **e** Relative amount of AA remaining in bacterial culture supernatants following 24 h incubation in 100 ppm. AA-supplemented media. Data shown represent the mean ± standard deviation (one-way ANOVA with Sidak's multiple comparisons) for $n = 3$ biological replicates performed in technical triplicate for each bacterial strain. **f** Co-occurrence network visually illustrating the significant interactions between taxa in the gut model. **g**, **h** Representative growth curves and **i** carrying capacity ($k$) of bacteria in 0.25 mM AA and acetate-supplemented media. Data represent mean ± standard deviation (one-way ANOVA with Sidak's multiple comparisons) of $n = 3$ biological replicates performed in technical triplicate for each bacterial strain. **j**, **k** Temporal overlay graphs showing *A. muciniphila* and Enterobacteriaceae abundances in the gut model over time alongside predicted menaquinone (vitamin K2) biosynthesis-related pathway abundances. RA = relative abundance. ****$P < 0.0001$ and n.s. = not significant.

interspecies electron transfer[19] and encourage growth of important symbionts in a homolog-specific manner[20]. Supporting this, a near-perfect overlap was observed between MK7, MK8, MK11, MK12, and MK13 biosynthesis pathways and relative abundance of *A. muciniphila* over time in the simulated gut model (Fig. 3j). In contrast, relative abundance of Enterobacteriaceae was better able to explain changes in MK6, MK9, and MK10 biosynthesis pathways over time (Fig. 3k and Supplementary Data 6). A stratified breakdown of MKn pathway contribution showed that Bacteroidales was the top contributor in PC patients irrespective of treatment group (Fig. 4j and Supplementary Data 7), which is consistent with recent reports from other human cohorts and mammalian models[21,22]. However, Enterobacteriales and Verrucomicrobiales (solely represented by *A. muciniphila* in this study) disproportionately contributed to the increased pathway abundance of MK6-MK13 in ADT + AA patients in the same homolog-specific pattern that was observed in the simulated gut model during AA exposure (Figs. 3j and 4j and Supplementary Data 8). Further supporting an AA-mediated shift in metagenomic potential that preferentially supports the growth of *A. muciniphila*, biosynthesis- and catabolism-related pathways for L-threonine and UDP-*N*-acetyl-glucosamine (two essential growth-promoting factors for *A. muciniphila*[23]) were also favorably altered during AA exposure in the gut model (Supplementary Fig. 7 and Supplementary Data 6).

## Discussion

This study demonstrates that the oral PC treatment drug AA can reproducibly modulate patient-associated GI microbial communities through promoting the growth of *A. muciniphila*. Notably, this study: (i) provides insight into this microbiota–xenobiotic interaction and the potential mechanism of cross-feeding between various bacterial species that are clinically relevant to PC treatments, and (ii) elucidates that bacterially mediated enrichment of host vitamin K2 status might be an auxiliary mechanism by which AA elicits its pro-survival effects in castrate-resistant PC cohorts.

The unusually high levels of contact between AA and the patient GI microbiota (due to poor absorption of the drug[6]) provides rich insight on the effect of localized delivery of selective bacterial growth agents. Specifically, our findings suggest that colonic delivery of the conjugated acetate portion of AA is the likely factor responsible for enrichment of *A. muciniphila* in PC patients (Figs. 1c and 3g–i). The importance of site-specificity in regard to exogenously delivered acetate is emphasized by past work showing that distal, but not proximal, colonic acetate infusions can improve metabolic disorders in overweight men[24]. Supporting the observed linkage between acetate and *A. muciniphila*, short-chain fatty acid metabolism is thought to influence *A. muciniphila* growth in the GI microbiota[25]. Serum acetate and

*A. muciniphila* abundance in the gut have also been shown to be positively correlated[5], although general consensus has long suggested that *A. muciniphila* is responsible for this increase though producing acetate itself. Nonetheless, acetate-producing *Bifidobacterium animalis* LMG P-28149 orally supplemented to mice can increase fecal *A. muciniphila* abundance by over a 100-fold[26] and engraftment of *A. muciniphila* in a simulated intestinal mucosa environment is sustained alongside cointroduction with mucin-colonizing and acetate-producing *Bacteroides*, *Ruminococcus*, and *Coprococcus* strains[27]. Thus, it may be that *A. muciniphila* capitalizes on the multifunctional glyoxylate pathway (predicted to be higher in ADT + AA patients; Fig. 4a, b), which is both required for mucin degradation[28] and enables bacterial growth on acetate via bypassing of the two decarboxylation steps of the citric acid cycle[18]. In addition, *A. muciniphila* also possesses a functional cytochrome *bd*, which when experimentally cloned into a cytochrome-deficient *Escherichia coli* mutant strain shifted metabolism in favor of acetate and ultimately doubled the maximum growth yield under oxygen limiting conditions[29]—similar to what would be expected at the intestinal lumen–epithelium interface[30]. Altogether with our findings, it appears that high acetate concentrations can accelerate *A. muciniphila* growth by acting as a direct energy source as well as triggering mucin degradation-related gene expression. While this phenomenon remains uncharacterized in *A. muciniphila*, analogous observations in *Pseudomonas aeruginosa* have demonstrated the simultaneous liberation and consumption of acetate during mucin degradation in the lung[28].

The fact that various Enterobacteriaceae members did not receive a growth benefit from AA or acetate in pure culture but increased in the simulated gut model during AA exposure (exhibiting an ~48 h lag in growth behind *A. muciniphila*) suggests a potential for mutualistic cross-feeding—which in light of differential menaquinone biosynthesis capabilities, supports previous findings of *A. muciniphila* requiring the exogenous co-occurrence of complementary quinones (i.e., ubiquinone, menaquinones) due to an incomplete biosynthetic pathway[31]. Moreover, cleavage of host-derived mucin by various commensals (including *A. muciniphila*) liberates fucose, which has been shown to stimulate the growth of certain Enterobacteriaceae via activation of their two-component sensing system, FusKR[32]. Uniquely among mucin degraders though, *A. muciniphila* can largely outcompete other bacteria for fucose in the gut and yield 1,2-propanediol from its metabolism[33]. Supporting this, *A. muciniphila* contributed substantially to the predicted increase in relative abundance of fucose degradation pathways in AA-exposed gut model samples (Supplementary Fig. 7D, and Supplementary Data 6 and 8). These findings suggest a scenario whereby *A. muciniphila* supports the maintenance of Enterobacteriaceae growth and in return benefits from gaining access

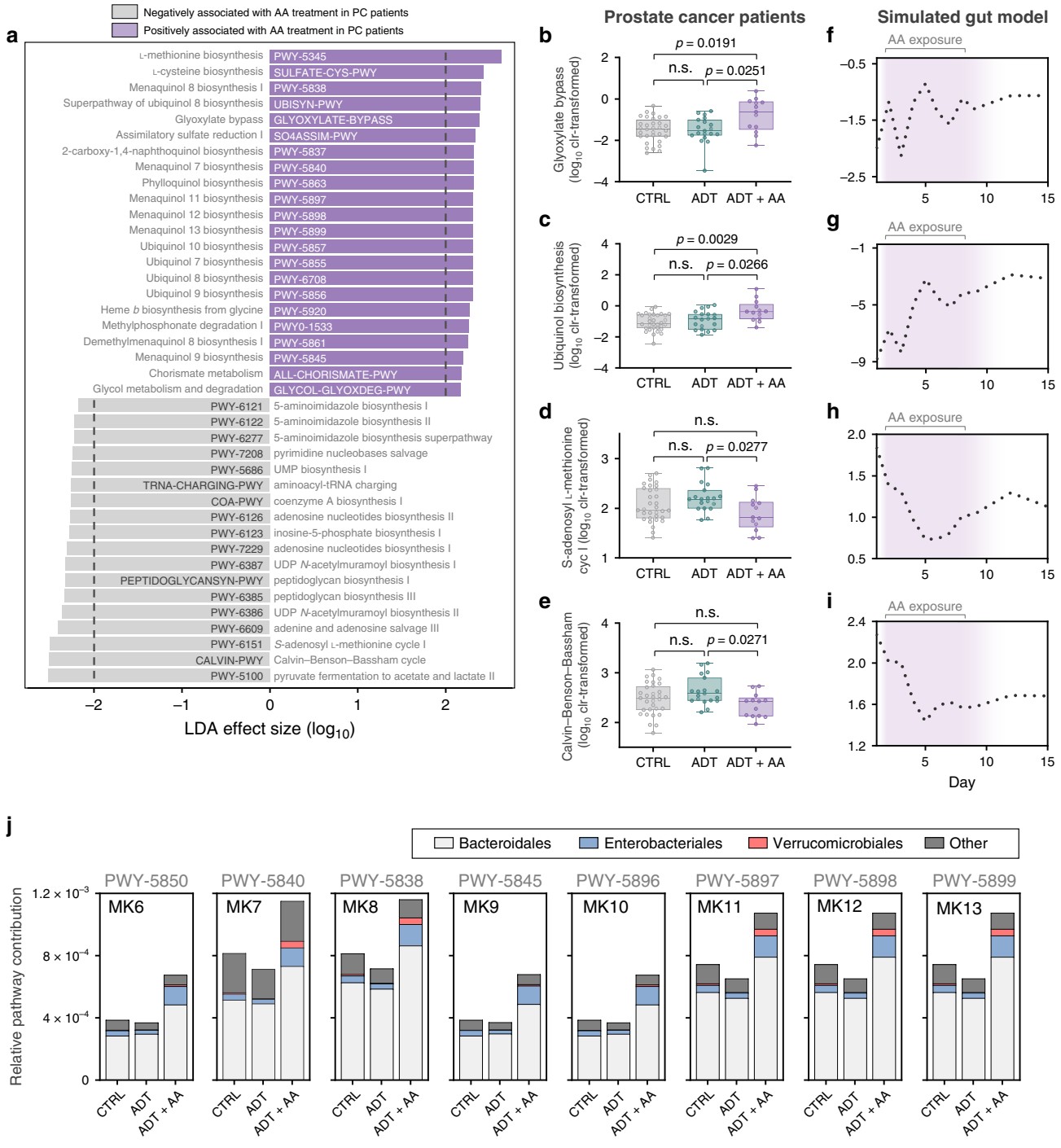

**Fig. 4 Abiraterone acetate (AA) exerts a reproducible effect on the human gut microbiota in vivo and in vitro. a** Differentially abundant pathways in patients receiving AA compared to those who were not. Multiclass analysis with the LEfSe algorithm was used to discriminate between AA effects overlapping systemic ADT exposure (LDA score >2 and p < 0.05 for all pathways shown). **b–e** Relative abundance of predicted pathways that were statistically increased in AA-treated patients. Data represent the median (line in box), IQR (box), and minimum/maximum (whiskers) of CTRL (n = 33), ADT (n = 21), and ADT + AA (n = 14) patient samples. Statistics shown for Kruskal–Wallis tests with multiple comparisons corrected using the Benjamini–Hochberg FDR method. **f–i** Relative abundance of predicted pathways in AA-exposed gut model samples. **j** Relevant bacterial contribution to each menaquinone biosynthesis pathway. Predicted pathways were inferenced using an exact amplicon sequence variant approach with the PICRUSt2 software and annotated using the MetaCyc metabolic pathway database. n.s. = not significant.

to a wider repertoire of quinones that cannot otherwise be synthesized, but that are required for optimal growth[20]. This might also attest to why *A. muciniphila* grows so poorly in pure culture (Fig. 3g–i).

The outer membrane protein Amuc_1100 can blunt tumorigenesis via modulation of CD8[+] cytotoxic T lymphocytes[4] and is

likely the most well-supported component of *A. muciniphila* for improving anti-PD1 immunotherapies. Here, we identify that *A. muciniphila*-mediated orchestration of communal menaquinone (i.e., vitamin K2) pools in the gut may represent an alternative mechanism by which this keystone species impacts cancer treatment outcomes. From the perspective of human biology,

vitamin K2 (dietary-derived or absorbed via microbial biosynthesis in the gut) acts as a ligand of the nuclear pregnane X receptor (also known as the steroid and xenobiotic sensing nuclear receptor, SXR, or NR1I2) and performs critical roles in blood coagulation, bone metabolism, and regulation of blood calcium levels and soft tissue calcification[34]. Interestingly, vitamin K2 is also a prospective anti-cancer agent that can target castrate-resistant PC in vitro[35], inhibit both androgen-dependent and androgen-independent tumor growth in mice[36], and is inversely associated with PC based on dietary intake results from a large European prospective study[37]. In-depth discussion on the physiological role of vitamin K2 is outside the scope of this study, although soft tissue calcification (a well-known consequence of vitamin K2 insufficiency) is frequently observed in a variety of diseased prostate specimens and is strongly correlated with PC[38].

Notably, *Corynebacterium* spp. (shown to be higher in control patients not receiving treatment; Fig. 1b) are expert colonizers of these calcified surfaces[39], more abundant in the urogenital tract of PC patients[40], and found at higher concentrations in prostate tumor and peri-tumor samples compared to non-tumor prostate tissue[41]. Coryneform bacteria are classically defined by the presence of unique α-branched β-hydroxy fatty acids called mycolic acids[42] (predicted biosynthesis pathways shown to be nearly abolished in both ADT and ADT + AA patients; Supplementary Fig. 8 and Supplementary Data 2 and 3) in their cell wall, which can evoke interleukin-23 signaling[43]—a key cytokine thought to contribute to tumorigenesis and progression to metastatic disease in the context of inflammation-related cancers[44]. While no clear etiological relationship has been established between *Corynebacterium* spp. and PC, it is interesting to speculate how androgen-inhibiting treatments as well as *A. muciniphila*-mediated modulation of quinone signaling and host vitamin K2 status may influence colonization of these common but potentially problematic bacterial species.

The seminal findings that steroidal progesterone derivate AA could inhibit CYP17A and thereby improve PC patient outcome in castrate-resistant cohorts has led to the widely accepted consensus that castrate-resistant PC remains a hormonally driven process[45]. Our results suggest that the efficacy of AA may be imparted through its ability to increase microbially synthesized vitamin K2 in PC patients via specific interactions with the key symbiont, *A. muciniphila*. Altogether, these findings expand on previous work from several other large studies that have demonstrated *A. muciniphila* to invoke a multiplicity of health benefits in their hosts and improve the efficacy of checkpoint inhibitor cancer immunotherapies[1–3]. We note that the prediction of underlying metabolic pathways was inferenced using an exact amplicon sequence variant (ASV) approach and that our findings should be further evaluated in a longitudinal intervention study design—preferably in conjunction with *A. muciniphila* and vitamin K2 treatment groups. Moreover, the direct inhibitory effects that abiraterone has on several human-derived steroid-metabolizing bacteria, as was identified in vitro in this study, also warrants further investigation.

## Methods

**Study design**. Patients were recruited as a sub-study of a Canadian Observational Study in Metastatic Cancer of the Prostate: a study of ZYTIGA use in the community urology setting (COSMiC; NCT02364531). The three PC patient groups examined included (1) patients not receiving any active treatment, (2) patients receiving ADT alone, and (3) patients receiving both ADT and treatment with orally administered AA. Additional information about the patients was collected, including age, body mass index, antibiotic use within 3 months, bicalutamide use, metformin use, oral steroid use, PSA level, radiation therapy exposure to pelvis, and whether or not they had metastatic disease (Supplementary Table 1). This study was approved by the Western University Research Ethics Board (IRB 00000940) and all participants (i.e., control PC patients, PC patients receiving ADT, PC patients receiving ADT + AA, and the healthy donor which provided a

sample for chemostat inoculation) were informed about the purpose of the study and signed a consent form prior to collection. Clinical samples were collected at Victoria Hospital (London, Ontario, Canada) between 2017 and 2018. Gloves used for digital rectal examinations of PC patients were collected and stored at 4 °C immediately afterwards for 1–2 days (a convenient storage period and temperature expected to be comparable to immediate freezing[46]). Subsequently, fecal matter was aseptically transferred from each glove using a single sterile polyester tipped swab (BD, Franklin Lakes, NJ) and stored in a 1.5 mL RNase/DNase-free microcentrifuge tube at −80 °C until DNA extraction.

**In vitro incubation of PC patient samples with AA**. A standardized inoculum of freshly collected fecal matter from male donors not receiving any form of PC treatment were homogenized in BHI media (catalog number: B11059, BD Difco) with the addition of 0.25 mM AA or vehicle (EtOH). After being incubated anaerobically at 37 °C for 24 h, debris was removed, and bacteria was harvested by centrifugation at $5000 \times g$ for 10 min and then stored at −80 °C until DNA extraction.

**Simulated model of the human gut microbiota**. The effect of AA exposure on human gut-associated bacterial communities was evaluated in a simulated human gut microbiota model using a Bioflo 110 Bioreactor (New Brunswick Scientific, NJ). Briefly, the bioreactor unit (4 L) was inoculated with freshly collected fecal matter from a healthy male donor and was allowed to stabilize over a period of 14 days prior to experimentation. An anaerobic state was maintained under continuous flow of nitrogen gas, a pH of 6.8 was maintained through daily titration with 0.5 M NaOH, and a constant input of established growth media relevant to the human intestinal environment was provided based on an established protocol for chemostat models of the human distal gut[47]. Following stabilization, one tablet of ZYTIGA (containing 250 mg AA) was dissolved in 5 mL ethanol and administered daily for several days, followed by a washout period (Fig. 3a). Effluent from the bioreactor was collected before, during, and after AA exposure and stored at −80 °C until DNA extraction.

**DNA extraction and 16S rRNA gene library preparation**. Patient rectal swabs, bioreactor effluent from the gut model, and in vitro incubated rectal samples were placed directly into the wells of a 96-well DNeasy PowerSoil HTP 96 kit (Qiagen) and the DNA extraction protocol was followed as per the manufacturer's instructions. Extracted DNA template was then transferred to a 96-well PCR plate. Targeted amplification of the 16S rRNA V4 region was performed using the established GOLAY-barcoded primers (5′–3′) ACACTCTTTCCCTACACGACG CTCTTCCGATCTNNNNxxxxxxxxxxxxGTGCCAGCMGCCGCGGTAA and (5′–3′) CGGTCTCGGCATTCCTGCTGAACCGCTCTTCCGATCTNNNNxxxxxx xxxxxxGGACTACHVGGGTWTCTAAT, wherein "xxxxxxxxxxxx" represents the sample-specific 12-mer barcode following the Illumina adaptor sequence used for downstream library construction[48]. Utilizing a BioMek Automated Workstation (Beckman Coulter), 2 µL of sample DNA (5 ng/µL) was added to a 96-well 0.2-mL PCR plate containing 10 µL of each primer per well (3.2 pmol/µL), followed by the addition of 20 µL of GoTaq 2X Colorless Master Mix (Promega). Final plates were then sealed using PCR-grade adhesive aluminum foil and placed in a Prime Thermal Cycler (Technie). PCR reaction conditions were as follows: an initial activation step at 95 °C, followed by 25 cycles of 95 °C for 1 min, 52 °C for 1 min, and 72 °C for 1 min. After completion, the thermocycler was held at 4 °C, and amplicons subsequently stored at −20 °C until further processing.

**16S rRNA sequencing and data analysis**. Processing of amplicon libraries was conducted at the London Regional Genomics Centre (Robarts Research Institute, London, Canada) in which amplicons were quantified using PicoGreen (Quant-It; Life Technologies, Burlington, ON), pooled at equimolar ratios, and sequenced on the MiSeq paired-end Illumina platform adapted for 2 × 250 bp paired-end chemistry. Sequence reads were then processed, aligned, and categorized using the DADA2 (v1.8) pipeline to infer exact amplicon sequence variants from amplicon data[49]. Patient rectal swab sample sequence reads were filtered (reads truncated after a quality score of ≤2 and forward/reverse reads truncated after 183/174 bases, respectively), de-replicated, de-noised, and merged using DADA2 default parameters with read recovery rates ranging from 72.3 to 93.8%. For the gut model and in vitro patient incubation samples, sequence reads were filtered (reads truncated after a quality score of ≤2 and forward/reverse reads truncated after 155/110 bases, respectively) and trimmed (10 bases off 5′ end of reverse reads) using optimized parameter settings as recommended. Next, sequence reads were de-replicated, de-noised, and merged using DADA2 default parameters with read recovery rates ranging from 88.3% to 91.8% for gut model samples and from 49.7% to 93.1% for incubated patient samples. Taxonomy was assigned to sequence variants using the SILVA non-redundant v132 training set. Contaminating taxa were identified and removed using frequency and prevalence methods in the *decontam* R package (v1.1.2) with PCR and environmental blanks as control samples. Outliers were identified within groups using the "codaSeq.outlier" function in the *CoDaSeq* R package (v0.99.4).

**Determination of diversity and differentially abundant taxa**. Pre-processing of sequence variant count matrices consisted of the following: zero imputation using the "cmultRepl" function (method = "CZM") in the *zCompositions* R package (v1.3.2-1), applying a per sample minimum abundance cutoff of 1%, and center log-ratio (clr) transformation of values using the "clr" method in the *compositions* R package. PCA was performed on genus-collapsed datasets with sample Aitchison distances (clr-transformed Euclidian distance) as input values using the "prcomp" function in R. Alpha and beta diversities for each sample were determined by calculating Shannon's *H* index and Aitchison's distance (within and between samples), respectively, using the QIIME2 software[50]. Statistical differences in alpha and beta diversities were determined by Kruskal–Wallis tests with Dunn's multiple comparisons. Differential abundance of genera were compared with the ALDEx2 tool[51], which uses clr-transformed posterior distribution of data generated from 128 Dirichlet Monte-Carlo instances to determine significantly different features within a compositional dataset[52]. Wilcoxon's rank-sum tests were used, followed by Benjamini–Hochberg false discovery rate (FDR) correction to identify genera that were significantly differed in relative abundance between groups using the "aldex.ttest" function of ALDEx2. In addition, effect size of differentially abundant taxa was calculated with the "aldex.effect" function of ALDEx2 since effect size measures have been shown to be more reproducible than *p* values[53].

**Correlations between metadata and microbiota variation**. Influencers of patient microbiota variation were identified by calculating the association between clinical metadata variables and genus-level community ordination (PCA based on clr-transformed Euclidian distance between samples) using the *envfit* function in the *vegan* R package (v2.5-6). Multivariate analysis of variance and linear correlations were performed on categorical and continuous variables, respectively, using the *envfit* function (999 permutation, significance set at FDR < 5%). Multivariate association testing was then performed to identify unique genus–covariate associations while de-confounding the effect of all other covariates. Patient features identified as significant covariates of microbiota variation (ADT, AA, and glucocorticoid usage) were used as predictor variables in boosted additive generalized linear models using the *MaAsLin2* (v0.99.1) R package. Genus-level abundances were clr-transformed, a minimum abundance cutoff of 1% was applied during pre-processing steps, and multiple comparisons were corrected with Benjamini and Hochberg FDR method.

**Co-occurrence network analysis of AA-exposed samples**. Genus-level correlations between taxa and AA exposure in the gut model were assessed by constructing a co-occurrence network using the *CoNet* software[54]. Briefly, ensemble inferencing with Pearson's and Spearman's correlations, Bray–Curtis dissimilarity, Kullback–Leibler divergence, and mutual information was used to create an initial association network. Subsequently, the edgeScores randomization function was used to perform 100 row-wise permutations with the 1000 highest and lowest scoring edges retained. The ReBoot renormalization function was then applied to address compositional bias and a merged final network was then constructed based on the score distribution of 100 bootstrap iterations. Significance was calculated using the Brown's method and multiple comparisons were corrected with the Benjamini–Hochberg FDR method. Edges that exceeded an adjusted FDR threshold of 1% were discarded. The network was visualized using an organic layout in Cytoscape (v3.7.2).

**Inferencing of microbial metagenomes**. Functional potential of microbial communities was determined by inferencing gene content from taxonomic abundances using an exact sequence variant approach with updated *PICRUSt2* software. *DADA2* processed ASVs were placed into a reference multiple-sequence alignment followed by sequence placement with the reference phylogeny database using the "q2-fragment-insertion plugin" in QIIME2. Subsequently, hidden-state predictions[55] were performed using the recommended maximum parsimony approach to predict gene family abundances and nearest-sequenced taxon index (NSTI) values were calculated. NSTI values are a metric used to identify ASVs that are highly distant from those that are available in the reference sequence database; high NSTI values are generally indicative of uncharacterized phyla or off-target sequences and are less informative. All ASVs from both patient and gut model samples were below the NSTI < 2.0 cutoff recommended for a high level of confidence in predictions. Unstratified metagenome predictions for EC numbers were normalized to 16S rRNA copies, regrouped to MetaCyc RXNs using the default mapping file, and then metabolic pathway abundances were inferenced with the MetaCyc pathway database[56]. The LEfSe algorithm was used with subclass analysis to discriminate between differentially abundant pathway associations with AA treatment and overlapping effects of concurrent ADT (combined LDA scores >2 and alpha values of <0.05 for factorial Kruskal–Wallis and pairwise Wilcoxon's tests were considered significant). In addition, effect size comparisons of differentially abundant pathways between individual patient groups were calculated using the ADLEx2 software[51].

**Quantitative PCR (qPCR)-based verification of bacterial loads**. Bacterial abundances of interest were confirmed by performing qPCR on DNA extracted from effluent of the gut model and patient fecal samples using established genus- and species-specific primer sets (Supplementary Table 6). Each 10 μl reaction was performed in technical triplicate using 2× SYBR Green PCR Master Mix (Applied BioSystems) following the manufacturer's instructions. Thermocycling conditions were as follows: 50 °C for 2 min, a cycle of 95 °C for 10 min, 40 cycles of 95 °C for 15 s, and 60 °C for 1 min. Subsequently, a dissociation melt-curve analysis was performed to assess amplification specificity: 95 °C for 15 s, 60 °C for 15 s, 0.075 °C/s increment increases in temperature until 95 °C. All qPCR reactions were performed in 384-well microplates on a QuantStudio 5 Real-Time PCR System (Applied Biosystems) and data were analyzed using the QuantStudio Design & Analysis Software v1.4 (Thermo Fisher Scientific).

**Bacterial growth assays**. *Akkermansia muciniphila*-type strain Muc (DSM 22959) was routinely cultured anaerobically (GasPak EZ container system; BD) at 37 °C using BHI (catalog number: B11059, BD Difco) supplemented with 5 μg/mL hemin (catalog number A11165; Alfa Aesar), 0.1 μg/mL vitamin K (catalog number 460-027-G005; Enzo), 0.125% [w/v] mucin (catalog number M2378; Sigma), and 0.5% [w/v] yeast extract (catalog number 212750; BD Difco, BHIS-YE). *Escherichia coli* O6:H1 CFT073 (ATCC 700928), *E. coli* 67[57], *E. coli* O18:K1:H7 UTI89[58], and the bacterial isolates derived from rectal swabs or the gut model (Supplementary Table 4) were routinely cultured at 37 °C under microaerophilic conditions. To determine the effect of AA on bacterial growth, freshly grown single colonies of bacteria were transferred to BHI-YE broth supplemented with either 0.25 mM AA, 0.25 mM sodium acetate, or vehicle (EtOH). For each strain, a minimum of three separate colonies were used with each colony subsequently being transferred in technical triplicate to a 96-well microplate (catalog no. 351177, Falcon) and incubated at 37 °C for 48–72 h. Growth was monitored intermittently by measuring optical density (OD) at a wavelength of 600 nm using a BioTek Microplate Spectrophotometer (Agilent). Summarization of growth characteristics and statistical analyses were performed using the *growthcurver* package (v0.3.0) in R[59].

**Determination of acetate utilization**. Bacterial strains isolated from the gut model (Supplementary Table 4) were qualitatively categorized based on their ability to utilize AA or sodium acetate as a sole carbon source (RCA + designation) or not (RCA – designation). Following an established protocol[60], bacterial isolates were streaked onto colorimetric selective media agar containing only sodium acetate as a sole carbon. A modified version of this assay, in which sodium acetate was replaced by an equimolar amount of AA, was also utilized to determine if the bacterial isolates could liberate bioavailable acetate from AA. Inoculated agar plates were then incubated under microaerophilic conditions at 37 °C and color change was evaluated at 24 and 48 h. Acetate utilization was also quantitively assessed for several strains of interest using M9 minimal media[61] containing 90.2 μM Na$_2$HPO$_4$, 22.0 μM KH$_2$PO$_4$, 8.6 μM NaCl, 9.3 μM NH$_4$Cl, 2 mM MgSO$_4$, and 0.1 mM CaCl$_2$. Overnight bacterial cultures grown aerobically in Luria-Bertani (catalog number: DF0446173, BD Difco) at 37 °C were washed thrice with 0.01 M phosphate-buffered saline, resuspended in M9 media supplemented with 0.25 mM AA or vehicle (EtOH), and then sub-cultured (1:40) into 96-well microplates (catalog no. 35177, Falcon). Plates were then incubated at 37 °C for 24 h in a BioTek Microplate Spectrophotometer (Agilent) with OD measurements at 600 nm taken every 30 min. Summarization of growth characteristics and statistical analyses were performed using the *growthcurver* package in R and GraphPad Prism (v8.3).

**Bacterial degradation of AA**. HPLC was used to quantify breakdown of AA in bacterial cultures. Briefly, bacterial strains of interest were grown overnight in media supplemented with 100 p.p.m. AA and then centrifuged (5,000 × *g*) for 10 min to obtain cell-free supernatant. Protein crash was performed by adding one volume of supernatant to one volume of HPLC-grade acetonitrile (catalog no. A996-4, Fisher), incubated at 4 °C for 15 min, and then centrifuged for 10 min (16,000 × *g*) at 4 °C. All samples were filtered using a 0.45 μm filter prior to HPLC analysis. AA (catalog no. SML1527, Sigma) standards were made in acetonitrile. All samples were analyzed using an Agilent 1100 HPLC instrument equipped with a degasser (G1379A), quaternary pump (G1311A), autosampler (G1313A), and diode array detector (G1315B). Samples were run on an Agilent Poroshell 120 EC-C18 (4.6 × 150 mm², 4 μm particle size) column with a Poroshell 120 EC-C18 (4.6 mm, 4 μm particle size) guard column kept at ambient temperature. The mobile phase consisted of an isocratic mixture of acetonitrile and HPLC-grade water (catalog no. W5-4, Fisher) (80:20 [v/v]) at a flow rate of 1.2 mL/min. The sample injection was 10 μL, and detection was performed at 254 nm. Run times were 10 min, with AA having a retention time of ~8.0 min. Data were analyzed using ChemStation A.10.02. The peak area of the samples was compared with the peak area of the external calibration curve to quantify AA.

**Reporting summary**. Further information on research design is available in the Nature Research Reporting Summary linked to this article.

## Data availability

Raw sequence reads were uploaded to the NCBI Sequence Read Archive and are accessible under BioProject ID PRJNA609050. All other remaining relevant data are

provided in the article, Supplementary information, or available from the corresponding author upon reasonable request.

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

## Acknowledgements

This study was in part funded by the W. Garfield Weston Foundation and by a Canadian Urologic Oncology Group (CUOG) grant. Abiraterone acetate was kindly provided by Janssen Pharmaceuticals.

## Author contributions

J.P.B., J.C., S.M.N., M.D., A.A., and G.R. conceived the study design. K.A.-R., A.A., M.D., S.M.N., and J.C. contributed to patient sample collection. B.A.D., R.M.C., K.A.-R., K.F.A., S.G., J.A.C., and H.W. performed laboratory experiments. B.A.D. and K.A.-R. drafted the manuscript and performed bioinformatic analyses. B.A.D., K.F.A., and J.P.B. contributed to figure preparation and interpretation of the data. All authors read, revised, and approved the final manuscript.

## Competing interests

The authors declare no competing interests.
