## [Peer Review File · Nature Communications]

REVIEWER COMMENTS

Reviewer #1 (Remarks to the Author):

This is an interesting paper highlighting a potentially novel mechanism of action for abiraterone (AA) via the gut flora. I am not really able to comment on the detailed analysis of the changes in flora beyond saying that the paper seems plausible but other reviewers will need to assess how the analytical work checks out. The authors show clear links between exposure to AA and changes in specific groups of gut floras as well as suggesting possible mechanisms for this via linked in vitro work. Additionally, secondary changes in Vit K levels offer a possible partial explanation for the efficacy of AA separate to the CYP17 pathway. The work appears to me to be novel and interesting as a clinician and I recommend publication.

Reviewer #2 (Remarks to the Author):

The manuscript by Daisley et al describes the enrichment of specific fecal bacteria in prostate cancer patients medicated by an androgen deprivation therapy (ADT) by gonadotropin-releasing hormone (GnRH) agonists and abiraterone acetate (AA). A case is made that *Akkermansia muciniphila* is enriched specifically. Moreover, *A. muciniphila* is among the enriched bacteria upon AA addition in a fermentor seeded with a healthy fecal inoculum. Finally, it is argued that *A. muciniphila* is capable of growth on AA via the acetate moiety. The authors conclude that their findings 'provide evidence that localized delivery of selective bacterial growth agents, via conjugation to insoluble substances, may offer a safer and more targeted solution'.

There are many scientific and technical issues with this manuscript and the most relevant ones are summarized below.

1. The authors show elegantly that *Corynebacterium* spp are decreased and *Akkermansia muciniphila* is increased on patients taking ADT or ADT + AA (Fig.1). However, the option that AA only shows this effect in the presence of ADT cannot be excluded since no patients with only AA mediation were studied. While it is possible that such patients do not exist, the conclusion that AA alone causes the observed effect is not supported by the experimental evidence.
2. The arguments that support the stimulation of *A. muciniphila* by AA and acetate are based on growth on BHI with small amounts of mucin (0.125 %) (Fig.2). This is unusual as *A. muciniphila* will produce acetate (and propionate) from mucin. It is likely that the amount of mucin used is too low as usually 0.5 % mucin is used. Moreover, acetate consumption is not shown. It may be possible that the potential need for acetate is an artefact of the peptide-rich BHI broth that is used as a base medium. The citations that are used to rationalize the need for acetate are either indirect or wrong. It has been reported that *A. muciniphila* produces more acetate and less propionate under microaerophilic conditions than without oxygen as a consequence of its capacity to use oxygen (not acetate) as electron sink! In the experiments shown (Fig.2G) one would like to see specific growth rate, acetate consumption and propionate production as well as the fate of the non-acetate part of AA as to support the conclusions.
3. As it stands now the authors discovered that in ADT + AA patients notably *A. muciniphila* is increased and they suggest this may be by the liberation of acetate from AA. Even if so, what does this mean? Is AA a complex way to deliver acetate in the gut? May colonic bacteria will use acetate, notably several butyrate producers such as *F. prausnitzii*. What are the pharmacokinetics of AA and how much will reach the colon? And if this occurs, what role would *A. muciniphila* have in the disease process? In absence of answers to those questions, the initial observations in the patients stands but the manuscript fails to provide a mechanistic explanation and the conclusion that a 'safer and more targeted solution' may be offered is much too premature. An intervention with *A. muciniphila* in a mouse (or other) model should be considered.

Other comments:

1. the fecal sampling of the patients was done in an indirect way. The M&M says 'Gloves used for digital rectal examinations of prostate cancer patients were collected and stored at 4°C immediately afterwards for 1-2 days'. This long time of incubation at 4 C may give a considerable bias in the community – samples should be processed right away or stored frozen at -80C.
2. The experiments with a cloned bd cytochrome gene in the cytochrome-deficient E. coli are wrongly cited: the used E.coli mutant will never make propionate but will make more acetate at the expense of lactate when it can respire with the cloned bd cytochrome
3. The authors describe a chemostat experiment but this is not a real chemostat as there is no growth limiting substrate and no steady state.
4. the use of PiCrust to infer functions should only be a last option: metagenomic analysis is to be preferred - all observations are really speculations that should be experimentally confirmed.

Reviewer #3 (Remarks to the Author):

The authors report the results of a study in which the effect of the administration of an oral drug treatment (AA) for prostatic cancer on the composition of the rectal fecal microbiota was determined by comparisons of PC patients. Taxonomic analysis based on the bacterial 16S rRNA gene revealed that two taxa were affected by drug treatment: *Corynebacterium* relative abundance was decreased (both ADT and ADT+AA therapies); *Akkermansia* relative abundance was increased (ADT+AA therapy). A direct action of AA on the proliferation of fecal bacteria in a chemostat was demonstrated in that the *Akkermansia* population was increased. These taxonomic changes were reflected in the prevalence of bacterial biosynthetic pathways, notably the synthesis of vitamin K2 associated with *Akkermansia*. The utilization of acetate derived from AA and growth of various bacterial strains belonging to species of interest were demonstrated in vitro. The authors conclude that treatment effects may, in part, be due to changes to rectal microbiota composition and that *Akkermansia* may be an indicator of successful treatment.

The manuscript is generally well prepared with clear Figures although axis labels in graphs need to be checked.

Suggestions for improving the manuscript:

The drugs are administered orally, therefore it can be supposed that they must pass to the rectum in efficacious amount to produce the alterations to microbiota composition. Has this been demonstrated? Wouldn't it be desirable that most of the drug is absorbed in the intestinal tract so that effective concentrations are delivered to the urogenital tract? So, please discuss the pharmacokinetics of ADT and AA in relation to your results. This would lead to an explanation of why *Corynebacterium* is affected by ADT, whereas the *Akkermansia* effect is produced by AA.

Please note that your statement "Our results suggest that the efficacy of AA may be imparted through its ability to directly inhibit potential pathobionts (e.g. *Corynebacterium* spp) ..." is incorrect. ADT therapy alters *Corynebacterium* abundance (Figure 1B). ADT+AA therapy does not produce a greater decrease.

Given that the urogenital tract of the patients is abnormal, could there be bacteria resident in the bladder that might be utilizing the drug or inhibited by it? You mention that *Corynebacterium* associates with PC tumor surfaces, so would it be more informative to examine the microbiota of urine rather than feces? There is a lack of rationale as to why the rectal feces were examined. It is

not sufficient to point to 'health benefits' of the gut microbiota – it seems that much of this work is not reproducible.

Perhaps as much as one half of the manuscript is about the chemostat experiment. It's a commendable approach and the comparison between in vivo and in vitro analyses is very interesting. However, the chemostat results seem to be based on a single fecal sample from a single healthy human and only one chemostat run was carried out. Is n=1 sufficient in this case?

It is suggested that Akkermansia could be used as a biomarker of treatment success, but PCA of microbiota data shows huge spread (overlap) within/between all three groups (CTRL, ADT, ADT+AA). So choosing one species as a biomarker may be risky if not impossible? Perhaps a cluster of bacterial species might be better – include cut-off levels in terms of relative abundances please.

Reviewer #4 (Remarks to the Author):

Daisley et al. present a well written and thorough exploration of the impact of ADT and ADT+AA prostate cancer treatments on the GI microbiota. ADT+AA treatment is shown to reduce the abundance of Corynebacterium species and enrich Akkermansia muciniphila in both in situ and host-free chemostat gut model in vitro samples. This study adds to the growing body of literature regarding the importance of the GI microbiota on the potential efficacy of therapeutics.

[Major]

- I do not believe the PCA plot in Fig. 1A supports the statement "... the microbiota composition of ADT and ADT+AA patients largely separated from CTRL patients along the PC1 axis and each other along the PC2 axis...". The 95% CI of ADT and ADT+AA have substantial overlap as do each of the treatments with the CTRL group.

- I'm unclear on what is present in Fig. 1D. Corynebacterium is indicated as having a median absolute log2 differences of around -4.3. Does this mean the median difference is around 0.05% as $\log_2(0.05) = -4.3$? Or is this plot still considering clr-transformed data? If so, it would help to indicate this directly in the figure and it would be helpful to have a plot (perhaps supplemental) indicating the non-transformed difference in relative abundance. I appreciate the mathematical advantages of clr-transformed data, but it is difficult to interpret what this means in terms of non-transformed differences in relative abundance and if this difference is large enough to be biologically relevant even if it is statistically significant.

- Figure 3 should state that gene content was inferred from the taxonomic profiles using PICRUSt2. I think it is important to be clear that gene content was not directly inferred using metagenomic sequencing.

[Minor]

- The manuscript is dense with abbreviations which is a barrier to readers that are not directly involved in this sub-discipline. I would recommend removing abbreviations that aren't commonly used or provide a substantial reduction in character/word count. For example, abbreviating control as CTRL seems unnecessary unless this is common practice in your sub-discipline, GnRH is provided as an abbreviation but never used again, CRPC is defined on line 56 but not used again until line 229 so perhaps should be restated in full on line 229.

- Reviewer comment
- Author Response

Reviewer #1 (Remarks to the Author):

This is an interesting paper highlighting a potentially novel mechanism of action for abiraterone (AA) via the gut flora. I am not really able to comment on the detailed analysis of the changes in flora beyond saying that the paper seems plausible but other reviewers will need to assess how the analytical work checks out. The authors show clear links between exposure to AA and changes in specific groups of gut floras as well as suggesting possible mechanisms for this via linked in vitro work. Additionally, secondary changes in Vit K levels offer a possible partial explanation for the efficacy of AA separate to the CYP17 pathway. The work appears to me to be novel and interesting as a clinician and I recommend publication.

We thank the reviewer very much for their positive feedback on our work. We are pleased to hear that others find the results as stimulating as we do.

Reviewer #2 (Remarks to the Author):

The manuscript by Daisley et al describes the enrichment of specific fecal bacteria in prostate cancer patients medicated by an androgen deprivation therapy (ADT) by gonadotropin-releasing hormone (GnRH) agonists and abiraterone acetate (AA). A case is made that *Akkermansia muciniphila* is enriched specifically. Moreover, *A. muciniphila* is among the enriched bacteria upon AA addition in a fermentor seeded with a healthy fecal inoculum. Finally, it is argued that *A. muciniphila* is capable of growth on AA via the acetate moiety. The authors conclude that their findings provide evidence that localized delivery of selective bacterial growth agents, via conjugation to insoluble substances, may offer a safer and more targeted solution.

There are many scientific and technical issues with this manuscript and the most relevant ones are summarized below.

1. The authors show elegantly that *Corynebacterium* spp are decreased and *Akkermansia muciniphila* is increased on patients taking ADT or ADT + AA (Fig.1). However, the option that AA only shows this effect in the presence of ADT cannot be excluded since no patients with only AA mediation were studied. While it is possible that such patients do not exist, the conclusion that AA alone causes the observed effect is not supported by the experimental evidence.

The reviewer is correct in their assumption that patient cohorts receiving only AA do not exist as systemic androgen deprivation therapy (ADT) is a starting point treatment for prostate cancers. We examined a broad range of patient variables during evaluation of potential confounders in the clinical metadata (Table S1). As stated on line 96, out of the 17 total factors evaluated, only AA ($p=0.001$), ADT ($p=0.043$), and corticosteroid treatment ($p=0.049$) had a significant impact on overall microbiota composition (Table S2). Thus, these findings suggest that no patient variables better explain the variance in microbiota composition than the PC treatments themselves and that ADT and AA have separately distinct influences on microbial communities in the gut. The use of multivariate-based analyses via generalized linear mixed models, as was implemented in this study using MaAsLin2 software (<http://huttenhower.sph.harvard.edu/maaslin2>), is a well-accepted and very common strategy employed in clinical studies to identify variables of interest responsible for an observed outcome or effect. (Gevers et al. 2014, Cell Host & Microbe; Jie et al. 2017, Nature Communications; Stewart et al. 2018, Nature; Yilmaz et al. 2019, Nature Medicine).

Nevertheless, we share the concern of the reviewer in regard to the background noise of biological systems and this is one of the reasons we performed several *in vitro* experiments as well as implemented the usage of an established host-free chemostat model of the human microbiota (McDonald et al. 2013, J Microbiol Meth). Using this this model, in which the confounding effects ADT therapy were absent, we demonstrated a near identical response in how human-associated microbial communities are altered upon exposure to AA. Moreover, inferencing of the metabolic potential also demonstrated remarkable overlap in the findings between the prostate cancer patient cohort and the chemostat model, which further supports the effects of AA are ADT-independent.

2. The arguments that support the stimulation of *A. muciniphila* by AA and acetate are based on growth on BHI with small amounts of mucin (0.125 %) (Fig.2). This is unusual as *A. muciniphila* will produce acetate (and propionate) from mucin. It is likely that the amount of mucin used is too low as usually 0.5 % mucin is used. Moreover, acetate consumption is not shown. It may be possible that the potential need for acetate is an artefact of the peptide-rich BHI broth that is used as a base medium. The citations that are used to rationalize the need for acetate are either indirect or wrong. It has been reported that *A. muciniphila* produces more acetate and less propionate under microaerophilic conditions than without oxygen as a consequence of its capacity to use oxygen (not acetate) as electron sink! In the experiments shown (Fig.2G) one would like to see specific growth rate, acetate consumption and propionate production as well as the fate of the non-acetate part of AA as to support the conclusions.

We do not discredit these facts and have cited literature directly pertinent to this topic on lines 210 (stating the positive association between serum acetate and *A. muciniphila*; Dao et al. 2016, Gut) and 212 (stating the influence of short chain fatty acids on growth of *A. muciniphila* growth in the gastrointestinal tract; Belzer and de Vos 2012, ISME J). Instead, we are simply suggesting that the produced acetate (or exogenously supplied acetate) may play an additional role in downstream metabolism of *A. muciniphila*. As far as we know, we are the first to report on the phenomenon in *A. muciniphila*. Notably, however, others have demonstrated a similar mechanism in *Pseudomonas aeruginosa* showing the simultaneous liberation and consumption of acetate during mucin degradation in the lung (Flynn et al. 2017, Infection and Immunity).

Our initial preliminary experiments identified that mucin precipitated at higher concentrations in BHI media. This was problematic for two reasons: i) it interfered with OD600 growth measurements of *A. muciniphila*, and most importantly ii) it was not amendable for AA breakdown assays via HPLC, as were performed in Fig. 2E.

Thus, to find an appropriate concentration of mucin to use, we serial diluted BHI + 0.5% mucin and then let the media stand overnight (~16 h) in an incubator at 37°C to emulate conditions for which it would be used

during downstream bacterial growth assays. Subsequently, we measured OD600 (as this was used to measure bacterial growth) and determined that a suitable concentration was 0.125% mucin. We chose this concentration on the basis that there was significantly less precipitation of mucin compared to the higher concentrations tested (indicated by a significantly higher OD600 measurement for BHI + 0.25% and BHI + 0.5% mucin samples) and no measurable difference in precipitation between the lower concentrations tested. Statistical analysis was performed on n=3 separate sets of serial dilutions using a one-way ANOVA with Tukey's multiple comparisons (all comparisons shown).

Beyond these preliminary experiments, 0.125% mucin concentration for growth of *A. muciniphila* is well supported by past literature. For example, in the original study first describing *A. muciniphila* (Derrien et al. 2004, Int J Syst Evol Microbiol), the authors did use a higher mucin concentration (0.25% [w/v]) to enrich for *A. muciniphila* from polymicrobial samples. However, later in the same study the authors used only 0.05% (w/v) mucin (derived from the HT-29 MTX human intestinal cell line) concentrations in their growth media during optimization and structural characterization experiments. Similarly, Guo et al. (2015, J Appl Microbiol) used 0.2% (w/v) mucin-supplemented media to cultivate isolates of *A. muciniphila* derived from a southern China cohort. Chia et al. (2018, Antonie Van Leeuwenhoek) used 0.125% (w/v) mucin and Wu et al. (2020, Anaerobe) used 0.1% (w/v) mucin, similar to the levels we used in the current study. Others have successfully grown *A. muciniphila* without the addition of mucin (Plovier et al. 2017, Nature Medicine; Henning et al. 2017, Anaerobe; Marcial-Coba et al. 2018, Food & Function; Seregin et al. 2017, Cell Reports; Zhang et al. 2018, Pathogens and Disease; Ring et al. 2018, Gut Microbes). In lieu of including our preliminary experiments as a supplementary file, we have added these necessary references to the methods section of the manuscript.

The rationale presented by the reviewer as to why a higher concentration of mucin would make a difference in growth (i.e. less acetate would be produced, presumably due to limited carbon) seems to only further support our novel viewpoint that acetate may be advantageous to the growth of *A. muciniphila* under a variety of conditions. Further supporting this, as well as addressing the reviewer's concern over peptide-richness of the BHI media used, the media used in our chemostat experiments contained both higher concentrations of mucin (0.4%) and human gut-realistic peptide levels (McDonald et al. 2013, J Microbiol Meth). Thus, a higher concentration of mucin and change in peptide richness, does not appear to negate the potent stimulatory effect of AA on *A. muciniphila*, as was demonstrated by an over 130-fold increase in *A. muciniphila* in response to AA in the chemostat model after only 24 h (Fig. 2B).

To further clarify this perspective, we present a 24-hour growth assay with *A. muciniphila* using 0.5% mucin media and compare it to our findings using 0.125% mucin media (following the same procedure detailed on lines 579-595 of the method section in the revised manuscript). After adjusting the optical density measurements to account for background noise of different mucin concentrations, we show there is indeed a trend ($P=0.1076$) towards increased growth of *A. muciniphila* in the vehicle BHI+0.5% mucin media compared to the vehicle BHI+0.125% mucin media. Importantly, these results are consistent with our original analyses and show a significant increase in growth of *A. muciniphila* in response to AA using both the BHI+0.125% mucin media ($P<0.0001$) and the BHI+0.5% mucin media ($P<0.0001$).

Tukey's multiple comparisons test		Mean Diff.	95.00% CI of diff.	Significant?	Summary	Adjusted P Value	
BH+0.125% muc-vehicle vs. BH+0.125% muc-100ppm AA		-0.3137	-0.3813 to -0.2460	Yes	****	<0.0001	A-B
BH+0.125% muc-vehicle vs. BH+0.5% muc-vehicle		-0.05617	-0.1238 to 0.01147	No	ns	0.1076	A-C
BH+0.125% muc-vehicle vs. BH+0.5% muc-100ppm AA		-0.2820	-0.3496 to -0.2144	Yes	****	<0.0001	A-D
BH+0.125% muc-100ppm AA vs. BH+0.5% muc-vehicle		0.2575	0.1899 to 0.3251	Yes	****	<0.0001	B-C
BH+0.125% muc-100ppm AA vs. BH+0.5% muc-100ppm AA		0.03167	-0.03597 to 0.09930	No	ns	0.4804	B-D
BH+0.5% muc-vehicle vs. BH+0.5% muc-100ppm AA		-0.2258	-0.2935 to -0.1582	Yes	****	<0.0001	C-D

Test details		Mean 1	Mean 2	Mean Diff.	SE of diff.	n1	n2	q
BH+0.125% muc-vehicle vs. BH+0.125% muc-100ppm AA		0.1953	0.5090	-0.3137	0.02112	3	3	21.00
BH+0.125% muc-vehicle vs. BH+0.5% muc-vehicle		0.1953	0.2515	-0.05617	0.02112	3	3	3.761
BH+0.125% muc-vehicle vs. BH+0.5% muc-100ppm AA		0.1953	0.4773	-0.2820	0.02112	3	3	18.88
BH+0.125% muc-100ppm AA vs. BH+0.5% muc-vehicle		0.5090	0.2515	0.2575	0.02112	3	3	17.24
BH+0.125% muc-100ppm AA vs. BH+0.5% muc-100ppm AA		0.5090	0.4773	0.03167	0.02112	3	3	2.120
BH+0.5% muc-vehicle vs. BH+0.5% muc-100ppm AA		0.2515	0.4773	-0.2258	0.02112	3	3	15.12

The reviewer's concerns with the literature cited in the manuscript have been addressed in "Other comments: Q#2" below. To address the reviewer's request of incorporating more descriptive statistics in regard to our *A. muciniphila* time course growth assays, we have performed several additional analyses that have now been incorporated as supplementary material. In brief, we demonstrate that the doubling time of *A. muciniphila* growth is reduced by nearly an hour in response to AA exposure, whereas the intrinsic growth rate, carrying capacity, and area under the curve are all significantly increased.

There is undoubtedly always more that can be done to narrow down the exact molecular mode of action.

However, we feel that a full mechanistic study will be needed to do proper justice and that any further distraction from the very pertinent clinical findings would only act to dilute the valuable information presented in this study.

3. As it stands now the authors discovered that in ADT + AA patients notably *A. muciniphila* is increased and they suggest this may be by the liberation of acetate from AA. Even if so, what does this mean? Is AA a complex way to deliver acetate in the gut? May colonic bacteria will use acetate, notably several butyrate producers such as *F. prausnitzii*. What are the pharmacokinetics of AA and how much will reach the colon? And if this occurs, what role would *A. muciniphila* have in the disease process? In absence of answers to those questions, the initial observations in the patients stands but the manuscript fails to provide a mechanistic explanation and the conclusion that a safer and more targeted solution may be offered is much too premature. An intervention with *A. muciniphila* in a mouse (or other) model should be considered.

We agree with the reviewer that these are all very important questions to be asked. AA is highly insoluble and up to 55% of parent compound is estimated to be excreted in the feces unaltered (Acharya et al. 2013, Xenobiotica). Based on this, our opinion is that delivery of acetate to the colon via conjugation to a highly insoluble substrate (like abiraterone) is an effective means of modulating microbial communities in the gut. It also draws light to the fact that prostate cancer patients may be depleted of acetate producers (like *Bifidobacterium* spp.) that cross feed the butyrate producers (like *F. prausnitzii*) (Rios-Covian et al. 2015, FEMS Microbiol Lett) as mentioned by the reviewer. Intriguingly, in the chemostat model during AA exposure, we observed a near perfect overlay of when *Akkermansia* increases and when *Faecalibacterium* decreases, and vice versa following cessation of AA exposure (shown in Fig. S3B below).

It is unlikely the reduction in *Faecalibacterium* was due to a direct inhibitory effect of *A. muciniphila*, given that co-incubation of *A. muciniphila* and *F. prausnitzii* has been shown to result in syntrophic growth and production of butyrate (Belzer et al. 2017, mBio). Instead, it could be proposed that either AA is directly toxic to *Faecalibacterium* or that the growth advantage of AA is simply allowing *A. muciniphila* to outcompete other microbiota members. The nuances of compositional data make it near impossible to decipher this interaction with certainty. However, it is indeed an interesting observation and we have added it to the supplementary data (Fig. S3) to improve understanding of these interactions in future studies.

We believe the mechanism by which *A. muciniphila* could improve the health of prostate cancer patients is through the production of unique vitamin K2 isoforms that are likely deficient in the majority of patient cohorts based on findings from a large European prospective study (Nimptsch et al 2008, Am J Clin Nutr). This information is discussed in detail on lines 209-231 of the revised manuscript. We can appreciate the review's concern over recommending targeted solution too prematurely, and as such have amended the relevant statements throughout the revised manuscript to be more conservative.

Other comments:

1. the fecal sampling of the patients was done in an indirect way. The M&M says ???Gloves used for digital rectal examinations of prostate cancer patients were collected and stored at 4??C immediately afterwards for 1-2 days???. This long time of incubation at 4 C may give a considerable bias in the community ??? samples should be processed right away or stored frozen at -80C.

We agree with the reviewer that this is an important consideration in sample collection. The glove sampling method was preferred as it enabled a higher rate of study enrollment than if full stool samples would have been requested from patients. Although immediate processing or storage at -80 is the "gold standard", it is often not feasible for clinical studies with large recruitment, our previous work has demonstrated there to be no "perfect" method for preservation of fecal samples, and that intraindividual differences that occur throughout storage are minimal when comparing across a broad range of temperatures (-80°C to +32°C), preservative agents (RNAlater vs none), and storage time (0 -7 days) (Al et al. 2018, J Microbiol Meth). Instead, more variation often occurs in the downstream sample processing and thus we instead focused our efforts on ensuring the samples were processed in an identical manner, and importantly, all samples were processed at the same time. We used the well-established DNeasy PowerSoil kit (also used for the Earth Microbiome Project), which implements several standards of excellence including that of beat beating, in order to maximize consistent yield of quality DNA extracted from samples. Beat beating is particularly important as past findings show it to result in significantly better bacterial community structure representation than other methods (Yuan et al. 2012, PLoS One). To satisfy the reviewer's concern, we have now incorporated the relevant references in the methods section of the revised manuscript.

2. The experiments with a cloned bd cytochrome gene in the cytochrome-deficient *E. coli* are wrongly cited: the used *E. coli* mutant will never make propionate but will make more acetate at the expense of lactate when it can respire with the cloned bd cytochrome

We believe the reviewer is referring to line 192 of the original manuscript: “*A. muciniphila*-derived cytochrome bd shifts metabolism in favor of acetate (over propionate) which ultimately increases ATP generation potential”.

The article that is being referenced in this section (Ouwerkerk et al. 2016; Appl Environ Microbiol) states the following about the *A. muciniphila* bd cytochrome: “*Enhanced growth under aerated conditions was observed together with a marginal but significant shift toward a higher acetate-to-propionate ratio. As cytochrome bd uses oxygen as a final electron acceptor, resulting in the production of H₂O and NAD (Fig. 6), to maintain the NAD/NADH ratios, the extra NAD production needs to be balanced by additional NADH regeneration. Therefore, the metabolism might shift from propionate, where a net total of 2 NADH are oxidized to 2 NAD, toward acetate production, where 1 NAD is reduced to 1 NADH.*”

Accordingly, we believe that our original statements stand correct. The intent of this sentence was not to imply that the mutant *E. coli* expressing *A. muciniphila*-derived bd cytochrome developed the ability to produce propionate (which we believe was the reviewer’s major concern). Rather, that this model demonstrated that increased acetate production was responsible for the higher acetate-to-propionate ratio observed in *A. muciniphila* grown under aerated conditions. To remove any uncertainty in this discussion point, all instances of propionate have been removed from the revised manuscript.

3. The authors describe a chemostat experiment but this is not a real chemostat as there is no growth limiting substrate and no steady state.

We are unclear on the reviewer’s concern as there was both a stable steady state attained in the chemostat model (Fig 3) used in this study as well as several growth-limiting substrates including peptone extract, yeast extract, casein, wheat starch, mucin, and pectin . To curb confusion, we have revised the methods section to include reference to the article by McDonald et al. (2013, J Microbiol Meth), which offers a comprehensive list of reagents as well as Macfarlane and Macfarlane (2007; Curr Opin Biotech) for an excellent overview of the experimental details and usefulness of the model itself. The Bioflo 110 Bioreactor (New Brunswick Scientific, NJ) that was used in this study satisfies these criteria and was specifically designed to operate as a chemostat. Moreover, demonstrating its multipurpose, others have also recently used this model as a chemostat to demonstrate directed evolution of the Calvin-Benson-Bassham cycle in *Escherichia coli* (Gleizer et al. 2019, Cell).

4. the use of PiCrust to infer functions should only be a last option: metagenomic analysis is to be preferred - all observations are really speculations that should be experimentally confirmed.

Yes, this is an excellent point brought up by the reviewer. We would like to clarify that PICRUST₂ was used in this study, and not the original PICRUST (version 1) which is nearly a decade old and indeed has many fallacies. This is a very important distinction that we suspect may have been overlooked given the very recent release of PICRUST₂ (Douglas et al. 2020, Nature Biotechnology). For example, the improved PICRUST₂ is now capable of predicting function for exact 16S rRNA gene sequencing variants as a result of several improvements to both sequencing platform error reduction (i.e. MiSeq v3) as well as downstream sequence read denoising software such as DADA2 (Callahan et al. 2016, Nature Methods) which was used in the current study. These critical improvements have led to the phasing out of operational taxonomic units (OTU; generally with amplicon sequences clustered at 97% identity) and the systematic supersession of exact amplicon sequence variants (SVs), which are higher-resolution analogues of the traditional OTU (Callahan et al. 2017, ISME J). Accordingly, this allows SVs to be matched at 100% identity to their

representative genomic potential and is far more accurate/informative than past implementations of best estimate ‘guessing’ of genomic potential based on the consensus genomic profile of OTUs clustered at 97% identity.

Translating this in a meaningful way based on established taxonomic boundaries (Yarza et al. 2014, Nature Reviews Microbiology), it’s the difference between predicting (with accuracy) the metagenomic potential at the genus level (94.5-98.7% sequence identity) using OTUs (i.e. PICRUSt) and at the species level (>98.7% sequence identity) using SVs (i.e. PICRUSt2). Arguably, accuracy in predictions is likely extended to the strain level for many organisms given the capacity of DADA2 to detect single nucleotide polymorphisms within a given microbial community. In addition, the reference genome database for PICRUSt2 has expanded over 10-fold compared to its predecessor – enabling exact genome matching for the large majority of SVs present, especially for human-derived samples, which dominate most 16S rRNA gene databases. Nonetheless, for query SVs without exact database matches, hidden-state predictions were performed using stringent thresholds (<2.0) when calculating the nearest-sequenced taxon index (NSTI) values, which are a metric used to identify how distant an unmatched SV is from those that are available in the genomic reference sequence database. All NSTI values from both patient and chemostat samples were <2.0, which might not be surprising given that both sets of communities are composed of human-derived bacteria and suggests a very high level of confidence in predictions (Douglas et al. 2020, Nature Biotechnology).

We agree with the reviewer that there are surely limitations to metagenomic inferencing, just as there are limitations to any other methodologies – including that of shotgun metagenomic sequencing itself (e.g. sample cost, low read depths of taxonomic marker genes with shallow shotgun approaches, cross-species contig and scaffolding errors, etc.). However, in situations where full shotgun metagenomic sequencing is not feasible due to financial aspects of larger clinical studies, recently improved SV-based inferencing can provide thought-provoking information and greatly improve the potential for novel hypothesis generation. Likewise, we believe that future readers will appreciate our extended efforts in this regard. We have ensured that the structural phrasing throughout the revised manuscript with regard to inferencing is conservative, supported by the data, and in no way misrepresents or exaggerates the study findings.

Reviewer #3 (Remarks to the Author):

The authors report the results of a study in which the effect of the administration of an oral drug treatment (AA) for prostatic cancer on the composition of the rectal fecal microbiota was determined by comparisons of PC patients. Taxonomic analysis based on the bacterial 16S rRNA gene revealed that two taxa were affected by drug treatment: *Corynebacterium* relative abundance was decreased (both ADT and ADT+AA therapies); *Akkermansia* relative abundance was increased (ADT+AA therapy). A direct action of AA on the proliferation of fecal bacteria in a chemostat was demonstrated in that the *Akkermansia* population was increased. These taxonomic changes were reflected in the prevalence of bacterial biosynthetic pathways, notably the synthesis of vitamin K2 associated with *Akkermansia*. The utilization of acetate derived from AA and growth of various bacterial strains belonging to species of interest were demonstrated in vitro. The authors conclude that treatment effects may, in part, be due to changes to rectal microbiota composition and that *Akkermansia* may be an indicator of successful treatment.

The manuscript is generally well prepared with clear Figures although axis labels in graphs need to be checked.

We appreciate the reviewer’s positive feedback and have ensured that all axis labels are correct in the revised manuscript.

Suggestions for improving the manuscript:

The drugs are administered orally, therefore it can be supposed that they must pass to the rectum in efficacious amount to produce the alterations to microbiota composition. Has this been demonstrated? Wouldn't it be desirable that most of the drug is absorbed in the intestinal tract so that effective concentrations are delivered to the urogenital tract? So, please discuss the pharmacokinetics of ADT and AA in relation to your results. This would lead to an explanation of why *Corynebacterium* is affected by ADT, whereas the *Akkermansia* effect is produced by AA.

Thank you for the suggestions. The absorption of the drug is indeed desirable. The conjugation of abiraterone with acetate is intended to improve the solubility, however AA is still quite insoluble and approximately 55% of the parent compound is estimated to be excreted in the feces unaltered (Acharya et al. 2013, *Xenobiotica*). We have added details pertaining to the pharmacokinetics of AA on lines 57-60 of the revised manuscript. The mechanism by which *Corynebacterium* spp. are reduced by the systemic androgen depleting effects of ADT, as compared to the enrichment of *A. muciniphila* seen with AA, is discussed on lines 105-136 of the revised manuscript.

Please note that your statement "Our results suggest that the efficacy of AA may be imparted through its ability to directly inhibit potential pathobionts (e.g. *Corynebacterium* spp)" is incorrect. ADT therapy alters *Corynebacterium* abundance (Figure 1B). ADT+AA therapy does not produce a greater decrease.

We believe this statement to be justified given our *in vitro* findings in Fig. S3 and Fig. S5 demonstrates that AA can directly inhibit several *Corynebacterium* isolates. To the best of our knowledge, we are the first to demonstrate these interactions and propose on lines 110-119 of the revised manuscript that it is likely due to abiraterone-mediated cross-reactivity with bacterial steroid hydroxylases found in *Corynebacterium* spp. However, we can appreciate the reviewer's concern given that there was indeed no enhanced depletion of *Corynebacterium* spp. in the ADT+AA group compared to the ADT group alone – likely due to the substantial decrease in available binding sites as a result of the background ADT effects. To address this concern and reduce the potential for misinterpretation of our findings, we have removed the first half and revised the sentence as follows: "Our results suggest that the efficacy of AA may be imparted through its ability to increase microbially synthesized vitamin K2 in PC patients via specific interactions with the common symbiont, *A. muciniphila*". Later in the same paragraph on lines 250-259, we take a more conservative approach to reminding the reader of our novel *in vitro* findings by stating: "Moreover, the direct inhibitory effects that abiraterone has on several human-derived steroid metabolizing bacteria, as was identified *in vitro* for the first time in this study, also warrants further investigation".

Given that the urogenital tract of the patients is abnormal, could there be bacteria resident in the bladder that might be utilizing the drug or inhibited by it? You mention that *Corynebacterium* associates with PC tumor surfaces, so would it be more informative to examine the microbiota of urine rather than feces? There is a lack of rationale as to why the rectal feces were examined. It is not sufficient to point to "health benefits" of the gut microbiota it seems that much of this work is not reproducible.

While it is not unreasonable to hypothesize that AA might interact with the microorganisms that are presumptively found on prostate tumor surfaces, it is unlikely that AA would reach this target site. For example, a mass balance study using ¹⁴C-labelled AA showed that the parent compound was rapidly converted to abiraterone *in vivo*, and that AA was undetectable (<0.2 ng/mL) in >99% of clinical plasma samples examined (Acharya et al. 2012, *Xenobiotica*). A recent study, performed at higher resolution, also confirmed this and showed that AA is found at very low levels in the plasma and likely even more so in peripheral tissues (Bouhajib and Tayab 2019, *Clinical Drug Investigation*). In contrast, approximately 55% of an administered dose of AA is excreted in the feces (Acharya et al. 2012, *Xenobiotica*). Thus, rectal fecal samples were chosen for evaluation instead of urine as we hypothesized there would be substantially higher chances for microbe-drug interactions to occur, not only because the bacterial load along the intestinal tract

is much higher than the urinary tract, but also because estimated AA concentrations are much higher. To satisfy the reviewer’s concern and ensure the study rationale is more apparent to future readers, we have added statements of justification on lines 57-68 of the revised manuscript.

Perhaps as much as one half of the manuscript is about the chemostat experiment. It’s a commendable approach and the comparison between in vivo and in vitro analyses is very interesting. However, the chemostat results seem to be based on a single fecal sample from a single healthy human and only one chemostat run was carried out. Is n=1 sufficient in this case?

Although we understand the reviewer’s concern on this matter, we believe that our validation studies substantially address this topic. Before investigating the prostate cancer cohort dataset (n=68 patients) and the chemostat model (n=11 timepoints across 15 days), we first performed *in vitro* validation experiments using n=8 separately collected fresh prostate cancer patient stool samples (now presented in Fig. 2 in the revised manuscript). In doing so, we demonstrated that samples characterized by “high” initial Akkermansia loads (>0.01% relative abundance; n=4) incubated with 100 ug/mL AA led to a significant reduction in Aitchison distance between samples (i.e. composition of microbial communities became more similar when exposed to AA; Fig. 2B below) and a trend towards decreased Shannon’s H Index (a well-established metric for alpha diversity; Fig. 2A below). Both of these findings demonstrate how “high” Akkermansia samples were directly responding to AA exposure. In contrast, there was no change in either of these microbial diversity metrics when samples with “low” initial Akkermansia loads (<0.01% relative abundance; n=4) were incubated with 100 ug/mL AA (Fig S2A-B below). Nevertheless, exposure to 100 ug/mL AA led to a detectable increase of Akkermansia in 6/8 of samples evaluated (Fig 2C).

Importantly, each of these results are consistent with our other findings in suggesting that: a) when Akkermansia is found at detectable limits, AA exposure leads to a pronounced expansion in its relative abundance (even after only 48 h), and b) the presence and abundance of Akkermansia play a key role in how microbial communities immediately respond to AA.

While we believe the chemostat model contributed a substantial amount to the study, and justified findings from several other experiments that were performed, we appreciate the reviewer’s concern that the main figures appear quite focused on this one dataset. Accordingly, we’ve reorganized the manuscript by transferring some of the supplementary data into the main body (alongside added discussion points) in efforts to reflect a more accurate and balanced representation of the study findings. We thank the reviewer for this very helpful suggestion.

It is suggested that Akkermansia could be used as a biomarker of treatment success, but PCA of microbiota data shows huge spread (overlap) within/between all three groups (CTRL, ADT, ADT+AA). So choosing one species as a biomarker may be risky if not impossible? Perhaps a cluster of bacterial species might be better ??? include cut-off levels in terms of relative abundances please.

The reviewer brings up an excellent point. But in contrast to their interpretation of the data, the overlap between all three treatment groups, is the exact reason why we specifically recommend only *Akkermansia* as a biomarker and not a cluster of bacterial species. As the PCA plot demonstrates in Fig. 1A, there are only subtle differences in microbiota composition between the groups, with key drivers “pulling” the groups apart. To dissect these differences, we utilized ALDEx2 software (Fernandes et al. 2013, PLoS One) to evaluate the underlying compositional differences and determined that *Akkermansia* was the only feature identifiable as significant for AA treatment when implementing both p-value and effect size thresholds – a statistical test of robustness for microbiota studies recommended for the production of more reproducible findings than use of either statistic alone (Gloor et al. 2017, Front Microbiol).

As requested, we have included commentary on relative abundance thresholds on lines 141-145 as a provisional guide for future studies and practitioners to consider. We have also clarified the importance of *A. muciniphila* as a “driver” that is singularly associated with the altered microbiota variation between patient cohorts on lines 128-130 and between AA exposed and non-exposed chemostat samples on lines 200-202.

Reviewer #4 (Remarks to the Author):

Daisley et al. present a well written and thorough exploration of the impact of ADT and ADT+AA prostate cancer treatments on the GI microbiota. ADT+AA treatment is shown to reduce the abundance of *Corynebacterium* species and enrich *Akkermansia muciniphila* in both in situ and host-free chemostat gut model in vitro samples. This study adds to the growing body of literature regarding the importance of the GI microbiota on the potential efficacy of therapeutics.

This is an accurate assessment of the study and we appreciate the reviewer’s thoroughness in their evaluation of our work.

[Major]

- I do not believe the PCA plot in Fig. 1A supports the statement ?????? the microbiota composition of ADT and ADT+AA patients largely separated from CTRL patients along the PC1 axis and each other along the PC2 axis??????. The 95% CI of ADT and ADT+AA have substantial overlap as do each of the treatments with the CTRL group.

This is a good point and mostly an oversight in phrasing on our behalf. As explained in the responses to reviewer #3, the intended purpose of the PCA plot in Fig 1A was not to demonstrate complete separation, but instead to highlight the significant features responsible for the “pulling” effect between groups (i.e. *Akkermansia* and *Corynebacterium*) by overlapping the representative eigenvectors (shown as arrows) to illustrate the substantial variation in strength of associations (i.e. length of the arrows). Nevertheless, we removed the word “separate” along with any of its derivative meanings as requested and instead have focused on the strength of associations between bacterial abundances and treatment group. The revised section is as follows:

“Summarizing the dataset, a principal component analysis (PCA) exploring patient microbiota differences (based on Aitchison distances between samples) demonstrated that the microbiota composition of ADT and ADT+AA patients shifted slightly from that of the control group along the PC1 axis and each other along the PC2 axis – accounting for 20.4% and 15.7% of interpersonal microbiota variability, respectively (Fig. 1A). The two largest influencers driving these directional shifts, based on genus level ordination, were found to be Akkermansia and Corynebacterium (Fig. 1B-C)”.

We thank the reviewer for their helpful suggestions.

- I'm unclear on what is present in Fig. 1D. Corynebacterium is indicated as having a median absolute log₂ differences of around -4.3. Does this mean the median difference is around 0.05% as $\log_2(0.05) = -4.3$? Or is this plot still considering clr-transformed data? If so, it would help to indicate this directly in the figure and it would be helpful to have a plot (perhaps supplemental) indicating the non-transformed difference in relative abundance. I appreciate the mathematical advantages of clr-transformed data, but it is difficult to interpret what this means in terms of non-transformed differences in relative abundance and if this difference is large enough to be biologically relevant even if it is statistically significant.

We agree with the reviewer that data interpretation is a key aspect of knowledge translation.

All plots in the original manuscript were showing log₁₀ clr-transformed data. The calculations performed by the reviewer appear to be producing frequency or proportion values, rather than percent relative abundance. The median percent relative abundance of Akkermansia is 0.1092% (95% CI = 0.0349-0.5649%) for the no treatment control group, 0.1128% (95% CI = 0.0307-0.3040%) for the ADT only group, and 0.5118% (95% CI = 0.0944-2.4360%) for the ADT+AA group. Thus, Akkermansia is approximately ~5x higher in the gut of ADT+AA patients compared to either of the other groups not receiving AA. We have incorporated this information into the written results of the revised manuscript to provide a clear comparison that is more easily linked back to biological relevance.

Moreover, in favor of the reviewer's recommendations on data presentation, we now present the abundance data for bacterial genera of interest in Fig 1 as simply "relative abundance" in the revised manuscript. To ensure there is no confusion regarding the veracity of statistical analyses performed, we've also stated in the Figure 1 legend that all that statistical measures were performed on log₁₀ clr-transformed values and that relative abundances are only shown for illustrative purposes.

- Figure 3 should state that gene content was inferred from the taxonomic profiles using PICRUSt2. I think it is important to be clear that gene content was not directly inferred using metagenomic sequencing.

As requested, we have included the following statement in the legend of Figure 3: *"Predicted pathways were inferred using an exact sequence variant approach with PICRUSt2 software and then annotated using the Metacyc metabolic pathway database"*.

[Minor]

- The manuscript is dense with abbreviations which is a barrier to readers that are not directly involved in this sub-discipline. I would recommend removing abbreviations that aren't commonly used or provide a substantial reduction in character/word count. For example, abbreviating control as CTRL seems unnecessary unless this is common practice in your sub-discipline, GnRH is provided as an abbreviation but never used again, CRPC is defined on line 56 but not used again until line 229 so perhaps should be restated in full on line 229.

Thank you for these helpful suggestions. The intended purpose of the 'CTRL' abbreviation was to aid in continuity between text and figures, though we can see how it could unnecessarily distract future readers. All instances of 'CTRL' and 'GnRH' have been removed, and castrate-resistant prostate cancer (CRPC) is now re-defined on line 229 of the revised manuscript.

We thank the reviewers for their insightful comments and believe that their suggestions have helped improve the quality and clarity of the manuscript for future readers

REVIEWER COMMENTS

Reviewer #2 (Remarks to the Author):

The present manuscript is very much improved and most questions have been satisfactorily addressed. What remains is the claim about the vitamin K2 (menaquinone) pathway: as indicated this is based on PiCrust2 predictions - hence an experimental validation is required as this is an important claim of the paper. What are the most relevant bacteria that make up this menaquinone pathway? In absence of metagenome information, can the authors provide evidence for the abundance of the *menA* or other key genes in the menaquinone biosynthesis as has been reported by Quinn et al 2019 – see below the URL below.

<https://www.tandfonline.com/doi/full/10.1080/19490976.2019.1710092>

Reviewer #3 (Remarks to the Author):

Overall, the authors have made reasonable attempts to modify the manuscript in accordance with reviewer's' opinions. Dealing specifically with responses to review 3:

I do not think that you have really addressed the question as to the reliability of results from a single chemostat run that was inoculated with feces from a single healthy human. It is appreciated that the focus of the manuscript has been moved slightly more towards in vivo data, but the n= 1 (one run, one human) is not resolved.

With regard to recognizing differential clusters of species relative to a single 'biomarker', it seems unusual that different abundances of *akkermansia* are not reflected in altered abundances of other bacterial types, given the highly interactive functioning of the gut microbiota. Perhaps a search for novel enterotypes would be useful?

With regards to review 2 comment on the chemostat, you really have not responded appropriately. Strictly speaking you need to show that sufficient turnovers of reactor volume have been achieved and that a particular substrate is at growth-limiting concentration. You can do this relatively easily when dealing with the use of a carbohydrate because a one-off increase in substrate concentration will cause a boost in bacterial numbers, then a return to steady state. Very hard to do with a rich medium where you can't define all of the carbon and energy sources. Maybe chemostat is the wrong word to use. A continuous culture not necessarily in steady state?

Reviewer #4 (Remarks to the Author):

Thank you for your clear and concise answers to my previous concerns. I have no further recommendations.

Reviewer #2 (Remarks to the Author):

The present manuscript is very much improved and most questions have been satisfactorily addressed. What remains is the claim about the vitamin K2 (menaquinone) pathway: as indicated this is based on PiCrust2 predictions - hence an experimental validation is required as this is an important claim of the paper. What are the most relevant bacteria that make up this menaquinone pathway? In absence of metagenome information, can the authors provide evidence for the abundance of the menA or other key genes in the menaquinone biosynthesis as has been reported by Quinn et al 2019 – see below the URL below. <https://www.tandfonline.com/doi/full/10.1080/19490976.2019.1710092>

Thank you for the suggestions. We have provided a visual representation of the bacterial contributions for each relevant menaquinone pathway in Fig. 4J of the revised manuscript. Similar to the findings in the linked article by Quinn et al. 2019, the major contributors to menaquinone production are members of Bacteroidales, Enterobacteriales, and Verrucomicrobiales. Furthermore, supporting our chemostat findings (Fig. 2J-K), Verrucomicrobiales (solely represented by *A. muciniphila*) disproportionately contributed to MK7, MK8, MK11, MK12, and MK13 pathways in ADT+AA patient samples with minimal influence on MK6, MK9, and MK10 pathways. As requested, a complete breakdown table for pathway contributions of bacteria at the genus level is provided in Table S13 and Table S14 for PC patient samples and the chemostat model, respectively.

Reviewer #3 (Remarks to the Author):

Overall, the authors have made reasonable attempts to modify the manuscript in accordance with reviewers' opinions. Dealing specifically with responses to review 3:

I do not think that you have really addressed the question as to the reliability of results from a single chemostat run that was inoculated with feces from a single healthy human. It is appreciated that the focus of the manuscript has been moved slightly more towards *in vivo* data, but the $n=1$ (one run, one human) is not resolved.

It should be noted that the chemostat model was not used as the sole means on which the study conclusions were drawn. Conversely, we utilized this convenient model as a form of secondary validation after already confirming our initial findings from the PC patient cohort by performing *in vitro* experiments with $n=8$ separately collected fresh PC patient stool samples. Given the high level of consistency across all sets of findings (patient data, *in vitro* culture data, and chemostat gut model data), we do not believe that further experimental replication using this model is necessary, nor would it be likely to provide any novel insight. We agree that, similar to most other model systems, there are limitations to the established human gut model that was used in our study and have cited relevant literature (McDonald et al. 2013, J Microbiol Meth) pertaining to these limitations on line 517. Nonetheless, we believe that the time-course dataset (consisting of 11 samples collected over a 15-day period) generated using the chemostat model will be helpful to future readers by providing an informative account of the temporal effects that AA can have on human gut-associated microbial communities.

With regard to recognizing differential clusters of species relative to a single 'biomarker', it seems unusual that different abundances of akkermansia are not reflected in altered abundances of other bacterial types, given the highly interactive functioning of the gut microbiota. Perhaps a search for novel enterotypes would be useful?

There was no evidence for novel enterotypes in this study, though we did identify changes in several taxa which co-occurred with *A. muciniphila*, namely members of Clostridia and Enterobacteriaceae (Fig. 3, Fig. 4). A variety of *Escherichia*, *Klebsiella*, and *Citrobacter* spp. were investigated *in vitro* (Fig. S2, Fig. S5, Table S5). These findings are discussed on lines 173-181 of the revised manuscript, and the overall importance summarized on lines 238-240. The reason why *A. muciniphila* is focused on as the 'biomarker' of interest is because the presence and abundance of this species was found to be the sole determinant driving AA-mediated modulation of microbial communities (as shown in Fig 2). Moreover, *in vitro* findings showed *A. muciniphila* was unique in that it was the only species that gained a growth advantage from AA in pure culture and in the presence of other carbon sources (Fig. 3G-I, Fig. S2, Fig. S5).

With regards to review 2 comment on the chemostat, you really have not responded appropriately. Strictly speaking you need to show that sufficient turnovers of reactor volume have been achieved and that a particular substrate is at growth-limiting concentration. You can do this relatively easily when dealing with the use of a carbohydrate because a one-off increase in substrate concentration will cause a boost in bacterial numbers, then a return to steady state. Very hard to do with a rich medium where you can't define all of the carbon and energy sources. Maybe chemostat is the wrong word to use. A continuous culture not necessarily in steady state?

The reviewer is correct that the continuous culture device is not strictly speaking in steady state, but rather is a dynamic model with a "stabilized" microbial community that emulates that of the human gut microbiota. To address this concern, we have replaced all instances of the word "chemostat" with either "dynamic human gut model" or "simulated human gut model".

We thank all reviewers for their comments, criticisms, and thoughtful responses, which have helped to improve both the quality and clarity of this manuscript.

REVIEWERS' COMMENTS:

Reviewer #2 (Remarks to the Author):

My earlier comments were very clear: ...' hence an experimental validation is required as this is an important claim of the paper.'

The authors give a visual representation of inputted metagenome information but no experimental validation: this is what was asked for - I even provided suggestion on how to do a simple qPCR analysis of the Akkermansia menA gene as has been done previously.

Reviewer #3 (Remarks to the Author):

The authors have made satisfactory modifications to the manuscript in accordance with reviewer comments.

- Reviewer comment
 - Author Response
-

Reviewer #2 (Remarks to the Author):

My earlier comments were very clear: ...' hence an experimental validation is required as this is an important claim of the paper.' The authors give a visual representation of inputted metagenome information but no experimental validation: this is what was asked for - I even provided suggestion on how to do a simple qPCR analysis of the *Akkermansia menA* gene as has been done previously.

We thank the reviewer for their recommendation. In the previously article referred to by the reviewer (Quinn et al. 2019, *Gut Microbes*), the authors used species-specific primer sets to measure copy number of *menA* in pathogen free mice with known microbiota compositions. Such an approach would not be amenable to usage in human gut-derived samples given the extensive taxonomic diversity and potential for non-specific primer binding between species sharing highly similar *menA* gene sequences.

More importantly, qPCR-based quantification of *menA* copy number would be ineffective at providing experimental validation as: 1) the presence/abundance of *menA* does not provide clear information regarding its functional expression or activity, and 2) *menA* encodes only a single enzyme involved in the production of menaquinone (Hiratsuka et al. 2008, *Science*). We discuss this later point several times throughout the manuscript in relation to the fact that *A. muciniphila* possesses an incomplete menaquinone biosynthesis pathway and thereby relies on the exogenous co-occurrence of menaquinones for optimal growth (Ravcheev et al. 2016, *Frontiers in Microbiology*).

Thus, while the reviewer's intent is appreciated here, we believe that the complex interactions between menaquinones, acetate, *A. muciniphila*, and the greater microbial community within the human gut microbiota is deserving of a complete investigation in future studies. However, to address the reviewer's original concern over experimental validation, we have provided an additional limitation statement in the final paragraph of the Discussion section as follows: "*We note that the prediction of underlying metabolic pathways was inferred using an exact amplicon sequence variant approach and that our findings should be further evaluated in a longitudinal intervention study design*".

Reviewer #3 (Remarks to the Author):

The authors have made satisfactory modifications to the manuscript in accordance with reviewer comments.

We thank all reviewers for their comments, criticisms, and thoughtful responses, which have helped to improve both the quality and clarity of this manuscript.